# Varying Shades of Wrong: Aligning LLMs with Wrong Answers Only

**Jihan Yao**[*1]  **Wenxuan Ding**[*2]  **Shangbin Feng**[*1]  **Lucy Lu Wang**[1,3]  **Yulia Tsvetkov**[1]

[1]University of Washington  [2]The University of Texas at Austin  [3] Allen Institute for AI

{jihany2, shangbin}@cs.washington.edu  wenxuand@utexas.edu

## Abstract

In the absence of abundant reliable annotations for challenging tasks and contexts, how can we expand the frontier of LLM capabilities with potentially wrong answers? We focus on two research questions: (1) *Can LLMs generate reliable preferences among wrong options?* And if so, (2) *Would alignment with such wrong-over-wrong preferences be helpful?* We employ methods based on self-consistency, token probabilities, and LLM-as-a-judge to elicit wrong-over-wrong preferences, and fine-tune language models with preference optimization approaches using these synthesized preferences. Extensive experiments with seven LLMs and eight datasets demonstrate that (1) LLMs *do* have preliminary capability in distinguishing various shades of wrong, achieving up to 20.9% higher performance than random guess; (2) Alignment with wrong-over-wrong preferences helps LLMs to produce less wrong and sometimes even outright correct answers, while improving overall model calibration. Code and data are publicly available at https://github.com/yaojh18/Varying-Shades-of-Wrong.

## 1 Introduction

Post-training with preference optimization, a.k.a. *alignment*, has become a crucial part of the development of large language models (LLMs) (Touvron et al., 2023). From online alignment with PPO (Schulman et al., 2017), to recent developments in offline alignment with DPO (Rafailov et al., 2023), LLMs improve by learning nuanced distinctions and separability between answers: a correct answer is preferred over an incorrect answer in reasoning problems (Wang et al., 2023c), or a response that adequately completes a user instruction is preferred over a failing one (Dubois et al., 2023), etc. This response "correctness" in alignment procedures and implementations typically comes from datasets with ground truth answers (Wang et al., 2023c), human annotation (Ouyang et al., 2022), or state-of-the-art LLMs (Lee et al., 2023), which are employed to generate high-quality reference answers to construct preference pairs and/or reward model training data.

However, in this work, we ask an emerging problem: *what if there is no correct answer in the alignment process*? What if the task comes without annotated ground-truths, is prohibitively expensive and time-intensive for human annotation, and even state-of-the-art LLMs are too poor to consistently provide correct answers? We see an increasing number of challenging benchmarks such as theorem proving (Welleck et al., 2021; Zhou et al., 2024) and structured reasoning (Ding et al., 2023; Fang et al., 2024), where absolute correct answers are difficult or impossible to collect. As a result, existing alignment procedures relying on high-quality expert-annotated data might struggle to expand the frontier of model capabilities. In response, we focus on the massive low-quality or even completely wrong answers generated by LLMs and propose *wrong-over-wrong alignment*, where LLMs align by learning to prefer less-wrong answers over more-wrong ones given the spectrum of wrongness. We present the outline of our work in Figure 1:

RQ1: *Can LLMs discriminate between varying shades of wrong and produce wrong-over-wrong preferences?* We experiment on four tasks with clear wrongness distinctions, e.g., to find the shortest path in a network with a ground truth path length of 5, finding a path of length 8 is "less wrong" than a path of length 11 (Wang et al., 2023a). For each task, we sample multiple solutions to a

---

*equal contribution

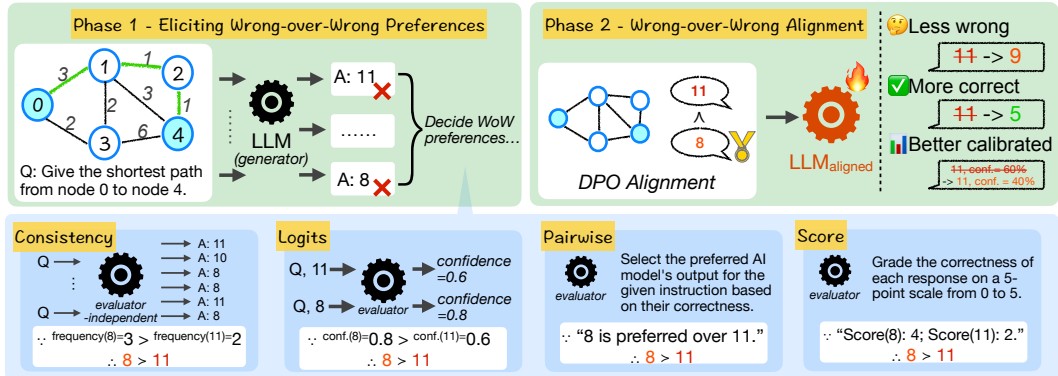

Figure 1: Two phases of aligning LLMs with wrong answers: *eliciting wrong-over-wrong preferences* and *wrong-over-wrong alignment*. In Phase 1, we employ four methods to elicit wrong-over-wrong preferences, based on answer consistency, logits-based confidence, and LLM-as-a-judge approaches. In Phase 2, we align LLMs with wrong-over-wrong preferences using DPO and expect to have less wrong, more correct, and better-calibrated answers.

problem, filter for incorrect answers, and employ LLMs to elicit wrong-over-wrong preferences via self-consistency (Wang et al., 2023d), token probabilities, pairwise comparison (Sun et al., 2024a), and LLM-as-a-judge(Dubois et al., 2023). We employ a quantitative "proxy"[1] of wrongness in each task to evaluate the quality of generated wrong-over-wrong preferences.

RQ2: *Is LLM alignment with wrong-over-wrong preferences helpful?* We fine-tune language models to align them using the LLM-generated wrong-over-wrong preferences. We hypothesize that wrong-over-wrong alignment will make models better calibrated, learning a better representation space for correct and incorrect responses. The models might produce more correct answers as determined by ground truth[2], or even if the answers are still incorrect, they will be less wrong based on the proxy.

Extensive experiments with seven open and proprietary models across eight datasets demonstrate the potential of wrong-over-wrong alignment: (1) LLMs *do* have preliminary capabilities to provide wrong-over-wrong judgements, with an accuracy of up to 70.9% across the four datasets spanning knowledge, factuality, commonsense, and structured data. We further identify that score-based methods together with margin filtering might be the best approach for preference elicitation and pairs of wrong answers with the *larger* gaps in scores should be retained. (2) Alignment with the elicited wrong-over-wrong preferences confirms the above hypotheses, demonstrating improvements across all three objectives: aligned LLMs are on average across datasets up to 9.0% less wrong as evaluated by the proxy, up to 7.0% of answers become correct post-alignment, while the Estimated Calibration Error (ECE) is reduced by up to 9.4% across the four tasks. Wrong-over-wrong alignment focuses on generating and selecting alignment data, functioning as an orthogonal complement to existing alignment algorithms. It demonstrates the unique potential to expand the frontier of model capabilities for tasks with clear wrongness distinctions and unavailable ground-truths.

## 2 METHODOLOGY

### 2.1 ELICITING WRONG-OVER-WRONG PREFERENCES

When LLM-generated answers are incorrect, it is often possible that some answers are *less wrong* than others. We term this "varying shades of wrong" and investigate whether LLMs could provide reliable wrong-over-wrong preferences if no reliable ground truths are available.

Formally, given a question $q$ and a pair of *wrong* answers $(a_1, a_2)$, we aim to employ LLMs to approximate a "ground-truth" wrong-over-wrong preference function $f(a_1 \succ a_2 \mid q) \rightarrow \{1, 0, -1\}$, where 1 indicates that $a_1$ is less wrong and should be preferred over $a_2$, $-1$ vice versa, and 0 in-

---

[1]Wrongness proxies are by no means *perfect*; they merely serve as objective and quantitative measures.

[2]While wrong-over-wrong alignment aims to improve models without ground-truths, ground-truths are still required to *evaluate* its effectiveness. Thus, we use datasets with ground-truths as examples in this paper.

dicates the two answers are not separable due to ambiguity or uncertainty. Since the ground-truth preference function $f(\boldsymbol{a}_1 \succ \boldsymbol{a}_2 \mid \boldsymbol{q})$ is usually unavailable, we propose using a silver function $\hat{f}(\boldsymbol{a}_1 \succ \boldsymbol{a}_2 \mid \boldsymbol{q})$ to estimate $f$ and employ $\hat{f}$ for evaluation (e.g., in the shortest path problem, a path of length 8 is less wrong than a path of length 11, when 5 is correct).

Although using proxy functions to directly construct wrong-over-wrong preferences may seem more ideal, these proxies often rely on the availability of ground-truth or models trained on ground truth. Moreover, not all datasets have well-defined proxies, and those that do vary vastly in format. Thus, we propose to elicit wrong-over-wrong preferences from an LLM, $f_{LLM}(\boldsymbol{a}1 \succ \boldsymbol{a}2 \mid \boldsymbol{q}) \to \{1, 0, -1\}$. Given that LLMs are known to suffer from poor calibration in evaluation (Liu et al., 2024), our first research question is: *Can LLMs provide reliable wrong-over-wrong preferences*, or can any $f_{LLM}$ be a good estimator of $\hat{f}$? We explore two typical LLM-as-a-judge methods:

**Pairwise comparison**    As increasingly employed in LLM-as-a-judge research (Dubois et al., 2023; Bai et al., 2023), an LLM can directly compare two answers. We prompt an LLM to reason about which answer in $(\boldsymbol{a}_1, \boldsymbol{a}_2)$ is less wrong (e.g., *"select the preferred AI model's output for a given instruction based on their correctness"* ). The result is denoted as $\mathrm{PC}(\boldsymbol{a}_1 \succ \boldsymbol{a}_2 \mid \boldsymbol{q}, \mathrm{LLM}) \to \{1, 0, -1\}$ and the full prompt text is in Table 13.

However, LLMs could be sensitive to answer order and suffer from positional biases (Wang et al., 2023b; Zheng et al., 2023). To mitigate this limitation, we employ consistency checks by flipping the order of answers, prompt the LLM to judge again, and disregard any inconsistent results. The final pairwise comparison preference function after filtering is:

$$f_{LLM}^{(p)}(\boldsymbol{a}_1 \succ \boldsymbol{a}_2 \mid \boldsymbol{q}) = \frac{1}{2}\big(\mathrm{PC}(\boldsymbol{a}_1 \succ \boldsymbol{a}_2 \mid \boldsymbol{q}, \mathrm{LLM}) - \mathrm{PC}(\boldsymbol{a}_2 \succ \boldsymbol{a}_1 \mid \boldsymbol{q}, \mathrm{LLM})\big)$$

**Score-based**    Another strategy is to let an LLM score the wrongness of each answer then compare the scores (Bansal et al., 2024). We employ the prompt *"grade the correctness of each response on a 5-point scale from 0 to 5"* to obtain the LLM's score for each answer $\boldsymbol{a}$: $\mathrm{score}(\boldsymbol{a} \mid \boldsymbol{q}, \mathrm{LLM})$. The full prompt text is in Table 14. This scoring mechanism allows for a more fine-grained measure than pairwise comparison. We identify two important factors in this scoring-based approach:

- **Batch size**: We simultaneously include $b$ answers $\{\boldsymbol{a}_i\}_{i=1}^{b}$ to the same question $\boldsymbol{q}$ for the LLM to score in one prompt. Increasing $b$ allows the LLM to compare and contrast multiple answers simultaneously, potentially improving its ability to discern relative answer reliability. There is a trade-off between efficiency/calibration and context length.

- **Score margin**: *Margin* indicates the absolute score difference between a wrong-over-wrong answer pair. Specifically, we use the $m$-percentile $M_m$ of all score margins to filter out pairs with smaller differences, as larger score margins are more trustworthy indicators of wrongness disparity (Li et al., 2024b; Yang et al., 2024; Yuan et al., 2024) (e.g., employing $M_{50}$ indicates that only the wrong-over-wrong pairs with the top 50% score gaps are retained.).

To sum up, the score-based preference function is defined as:

$$f_{LLM}^{(s)}(\boldsymbol{a}_1 \succ \boldsymbol{a}_2 \mid \boldsymbol{q}) = \mathrm{sgn}(\mathrm{score}_{diff}) \cdot \mathbb{1}(|\mathrm{score}_{diff}| > M_m))$$

where $\mathrm{score}_{diff} = \mathrm{score}(\boldsymbol{a}_1 \mid \boldsymbol{q}, \mathrm{LLM}) - \mathrm{score}(\boldsymbol{a}_2 \mid \boldsymbol{q}, \mathrm{LLM})$ and $\mathbb{1}(\cdot)$ is the indicator function.

For comparison, we also include three baseline methods:

**Heuristic**    Research suggests that response length correlates with answer quality (Zhao et al., 2024), i.e., longer answers with detailed reasoning steps may be closer to a correct solution. We define this trivial heuristic preference function as: $f_{noLLM}^{(h)}(\boldsymbol{a}_1 \succ \boldsymbol{a}_2 \mid \boldsymbol{q}) = \mathrm{sgn}(\mathrm{len}(\boldsymbol{a}_1) - \mathrm{len}(\boldsymbol{a}_2))$, where $\mathrm{sgn}$ is the sign function.

**Consistency-based**    For multiple sampled responses, the frequency of answers can be interpreted as a measure of model confidence (Wang et al., 2023d; Manakul et al., 2023; Miao et al., 2024). Given $m$ sampled answers $\{\boldsymbol{a}_i\}_{i=1}^{m}$, the sampling repetition score is calculated as $\mathrm{sr}(\boldsymbol{a}_i) = \frac{1}{m}\sum_{j=1}^{m}\mathbb{1}(\boldsymbol{a}_i = \boldsymbol{a}_j)$, representing how frequently an answer is repeated across multiple samples.

Based on the consistency scores, the preference function is defined as: $f_{noLLM}^{(c)}(\boldsymbol{a}_1 \succ \boldsymbol{a}_2 \mid \boldsymbol{q}) = \mathrm{sgn}(\mathrm{sr}(\boldsymbol{a}_1) - \mathrm{sr}(\boldsymbol{a}_2))$, assuming that the more frequent answers are more likely to be correct (Cole et al., 2023). Note that $f^{(c)}$ might not be applicable to open-ended generations where there is no fixed set of candidate answers.

**Logits-based** A well-calibrated LLM's token probabilities should reflect the reliability of responses (Geng et al., 2024), and a more probable answer is more likely to be correct. We employ Negative Log-Likelihood (NLL) to estimate the LLM's confidence in each wrong answer, specifically $\mathrm{NLL}(\mathbf{a} \mid \mathbf{q}) = -\log(\prod_{t \in \mathbf{a}} p_t) = -\sum_{t \in \mathbf{a}} \log(p_t)$ where $t$ is a token in answer $\mathbf{a}$ and $p_t$ is its token probability. The logits-based preference function is then defined as: $f_{LLM}^{(l)}(\boldsymbol{a}_1 \succ \boldsymbol{a}_2 \mid \boldsymbol{q}) = \mathrm{sgn}(\mathrm{NLL}(\boldsymbol{a}_2 \mid \boldsymbol{q}) - \mathrm{NLL}(\boldsymbol{a}_1 \mid \boldsymbol{q}))$.

## 2.2 WRONG-OVER-WRONG ALIGNMENT

We hypothesize that there are valuable signals in wrong-over-wrong preferences, that LLMs could learn to distinguish answer reliability, improve model calibration, and more. Wrong-over-wrong alignment is a set of synthetic experiments which provide a controlled environment to study under the worst scenario with no ground-truths available, whether we can still push the boundaries of model capabilities using only low-quality answers. Thus in practice, it is unnecessary to avoid right-over-wrong alignment. Instead, we show that wrong-over-wrong alignment is a good supplement to right-over-wrong alignment in Table 3.

To this end, we first synthesize a dataset $\mathcal{D}_{WoW}$ through the following pipeline. First, given an original dataset $\mathcal{D}_0 = \{(\boldsymbol{q}^{(i)}, \boldsymbol{a}_{gt}^{(i)})\}_{i=1}^n$, we sample $m$ answers from the LLM and synthesize scenarios with wrong answers only by filtering out correct ones: $\{\boldsymbol{a}_j^{(i)} \mid \boldsymbol{a}_j^{(i)} \neq \boldsymbol{a}_{gt}^{(i)}\}_{j=1}^m \sim \mathrm{LLM}(\boldsymbol{q}^{(i)})$. Then we iterate over every combination $(\boldsymbol{a}_j^{(i)}, \boldsymbol{a}_k^{(i)})$ of $\{\boldsymbol{a}_j^{(i)} \mid \boldsymbol{a}_j^{(i)} \neq \boldsymbol{a}_{gt}^{(i)}\}_{j=1}^m$, and judge the answers with $f_{LLM}$ or $f_{noLLM}$. The one considered less wrong is the chosen response $\boldsymbol{a}_c$ and the other is the rejected response $\boldsymbol{a}_r$. We ignore the cases where the LLM gives a tie ($f_{LLM} = 0$). Full details of wrong-over-wrong dataset construction are available in Algorithm 1.

With this dataset, we fine-tune an LLM using any preference optimization method (Azar et al., 2024; Hong et al., 2024; Wu et al., 2024; Calandriello et al., 2024). For example, the training objective of DPO can be expressed as:

$$\mathcal{L}_{DPO}(\pi_\theta; \pi_{ref}) = -\mathbb{E}_{(\boldsymbol{a}_c, \boldsymbol{a}_r, \boldsymbol{q}) \sim \mathcal{D}_{WoW}} \left[ \log \sigma \left( \beta \log \frac{\pi_\theta(\boldsymbol{a}_c \mid \boldsymbol{q})}{\pi_{ref}(\boldsymbol{a}_c \mid \boldsymbol{q})} - \log \frac{\pi_\theta(\boldsymbol{a}_r \mid \boldsymbol{q})}{\pi_{ref}(\boldsymbol{a}_r \mid \boldsymbol{q})} \right) \right]$$

where $\pi_\theta$ is the policy model and $\pi_{ref}$ is the reference policy. To evaluate our hypothesis about the benefits of wrong-over-wrong alignment, we measure model performance from three aspects:

**Less wrong** We evaluate the wrongness of answers that are still wrong post-alignment based on $\hat{f}(\boldsymbol{a}_1 \succ \boldsymbol{a}_2 \mid \boldsymbol{q})$. A reduction in wrongness indicates that LLMs are producing a better attempt at a challenging problem. As LLMs learn to separate varying shades of wrong in the alignment tuning process, we expect the answer distribution of LLMs to move towards less-wrong direction.

**More correct** We also evaluate the accuracy of models producing correct answers. An increase in accuracy demonstrates that surprisingly, training on wrong answers *only* can guide models to produce correct answers. As the model moves its output distribution to "less wrong" direction, answers that were previously close-to-correct may be adjusted enough to align with the correct solution, even though the training process only employs incorrect answers.

**Better calibrated** By training the model to prefer less-wrong answers, it learns to make finer distinctions between varying shades of wrongness and its confidence estimation should become more reliable. We evaluate the calibration of models based on token probabilities.

## 3 EXPERIMENT SETTINGS

**Models** We employ three open and proprietary LLMs for experiments spanning different scales and access levels. First, we use LLAMA3-8B (Dubey et al., 2024), GPT-3.5, and GPT-4O (Achiam

et al., 2023) to sample 10 answers per problem with random option orders for multiple-choice questions to increase the robustness of evaluation. We employ a temperature of 1.0 and a max generation length of 1024. LLMs employed in this stage are called *generators*. We then elicit wrong-over-wrong preferences using the same LLMs with each of $f_{noLLM}^{(h)}$, $f_{noLLM}^{(c)}$, $f_{LLM}^{(l)}$, $f_{LLM}^{(p)}$, and $f_{LLM}^{(s)}$. LLMs employed in this stage are called *evaluators*. In Appendix A, we also investigate the performance of GEMINI-FLASH, GEMINI-PRO (Team et al., 2023), MISTRAL-7B (Jiang et al., 2023), GEMMA-7B (Team et al., 2024) and aggregated evaluators.

Finally, we conduct QLoRA fine-tuning (Dettmers et al., 2023) on LLAMA3-8B using the collected wrong-over-wrong preferences through DPO. More preference optimization methods are experimented with in Appendix A. We experiment on preferences elicited from 4 methods, respectively $f_{GPT-4o}^{(p)}$ with consistency checks, $f_{GPT-4o}^{(s)}$ with $M_{50}$ and $M_{10}$, and $\hat{f}$ as an oracle reference. We sample 4000 wrong-over-wrong pairs for each experiment.

**Data** We employ datasets with a *carefully selected* but not necessarily *perfect* proxy correctness function $p(\boldsymbol{a} \mid \boldsymbol{q})$. Specifically, the proxy function gives a heuristic-based or model-based correctness score to each answer based on ground-truth and by comparing $p(\boldsymbol{a} \mid \boldsymbol{q})$ we can know which answer is "less wrong". To have more accurate evaluation and mitigate the bias and subjectivity brought by a single proxy, we include a wide range of proxies across various tasks:

- **Knowledge Crosswords (KC)** (Ding et al., 2023) is a multiple-choice structured knowledge reasoning benchmark where LLMs are tasked with filling three blanks with entities and satisfy given factual constraints. The correctness proxy is $p^{(kc)}(\boldsymbol{a} \mid \boldsymbol{q}) = \frac{\text{\# of correctly filled blanks}}{\text{\# of all blanks(3)}}$.

- **NLGraph (NLG)** (Wang et al., 2023a) is a graph reasoning benchmark and we employ the shortest path subset, where LLMs find the shortest path between two nodes in a weighted undirected graph. The correctness proxy is $p^{(sp)}(\boldsymbol{a} \mid \boldsymbol{q}) = 1 - \frac{|w_{\boldsymbol{a}} - w_{gt}|}{w_{worst} - w_{gt}}$, with $w_{\boldsymbol{a}}$ being the weight of the shortest path in $\boldsymbol{a}$, $w_{gt}$ being ground truth, and $w_{worst}$ being the weight of the longest simple path.

- **Bio Generation (BG)** LLMs are asked to generate a biography of a named individual, and the correctness of the answer is evaluated via FActScore (Min et al., 2023), which computes the percentage of atomic facts supported by retrieved passages. The names and retrieved passages are sourced from Wikipedia. The proxy function is defined as $p^{(bg)}(\boldsymbol{a} \mid \boldsymbol{q}) = \text{FActScore}(\boldsymbol{a} \mid \boldsymbol{q}) \in [0, 1]$.

- **COM$^2$** (Fang et al., 2024) is a multiple-choice commonsense reasoning benchmark. We use Vera (Liu et al., 2023) to obtain the commonsense plausibility scores for each option. The proxy function is defined as $p^{(cs)}(\boldsymbol{a} \mid \boldsymbol{q}) = \text{Vera}(\boldsymbol{a} \mid \boldsymbol{q}) \in [0, 1]$.

We sample 625, 625, 625, and 380 questions from each dataset, each split into training sets $\mathcal{D}_{train}$, validation sets $\mathcal{D}_{val}$, and test sets $\mathcal{D}_{test}$ with an approximately 8:1:1 ratio. We then generate $\mathcal{D}_{WoW}$ from $\mathcal{D}_{train}$ employing the pipeline in §2.2. $\mathcal{D}_{test}$ is used for evaluation and $\mathcal{D}_{val}$ for hyperparameter tuning. We also experiment on datasets without clear right-wrong distinctions in Appendix A. More detailed experiment settings can be found in Appendix B.

**Evaluation and Metrics** For eliciting wrong-over-wrong preferences, we evaluate by the proxy ground-truth preferences $\hat{f}(\boldsymbol{a}_1 \succ \boldsymbol{a}_2 \mid \boldsymbol{q})$. The accuracy of LLM-generated wrong-over-wrong preferences is defined as: $\text{Acc}_{WoW} = \frac{1}{|\mathcal{D}_{WoW}|} \sum_{(\boldsymbol{q},\boldsymbol{a}_c,\boldsymbol{a}_r) \in \mathcal{D}_{WoW}} \mathbb{1}(\hat{f}(\boldsymbol{a}_c \succ \boldsymbol{a}_r \mid \boldsymbol{q}) = f_{(no)LLM}(\boldsymbol{a}_c \succ \boldsymbol{a}_r \mid \boldsymbol{q}))$. For wrong-over-wrong alignment, we compare WoW-aligned models with unaligned base models and evaluate improvements across three objectives: less wrong, more correct, and better calibration.

- **Less Wrong**: We employ the average proxy function scores of all wrong answers: $p_{wrong} = \frac{1}{N_{wrong}} \sum_{\mathcal{D}_{test}} \sum_{\boldsymbol{a}_i \neq \boldsymbol{a}_{gt}} p(\boldsymbol{a}_i \mid \boldsymbol{q})$, where $N_{wrong}$ is the number of wrong answers and $\boldsymbol{a}_{gt}$ indicates the correct answer.

- **More Correct**: We employ task accuracy: $\text{Acc} = \frac{1}{N} \sum_{\mathcal{D}_{test}} \sum_{\boldsymbol{a}_i} \mathbb{1}(\boldsymbol{a}_i = \boldsymbol{a}_{gt})$, where $N$ is the number of all answers.

- **Better Calibration**: We extract the confidence of LLM answers with $\exp(\text{NLL}(\mathbf{a} \mid \mathbf{q})) \in [0, 1]$, create 10 bins of $[0.1i, 0.1(i + 1)]$, and employ Expected Calibration Error (ECE) (Guo et al., 2017) to measure the calibration quality.

| Method | Margin | LLaMA3-8B | | | | GPT-3.5 | | | | GPT-4o | | | | Overall |
|---|---|---|---|---|---|---|---|---|---|---|---|---|---|---|
| | | KC | BG | COM² | NLG | KC | BG | COM² | NLG | KC | BG | COM² | NLG | |
| **EVALUATOR-INDEPENDENT** | | | | | | | | | | | | | | |
| HEURISTIC | $M_{50}$ | .483 | .480 | .498 | .492 | .546 | .519 | .482 | .463 | .491 | .459 | .444 | .503 | .488 |
| | $M_{10}$ | .502 | .425 | .492 | .514 | .589 | .568 | .489 | .420 | .500 | .408 | .380 | .533 | .474 |
| CONSISTENCY | $M_{50}$ | .500 | - | .505 | .582 | .526 | - | .470 | .605 | .434 | - | .565 | .548 | .559 |
| | $M_{10}$ | .447 | - | .441 | .578 | .506 | - | .556 | **.695** | .600 | - | .423 | .494 | .566 |
| **LLAMA3-8B AS EVALUATOR** | | | | | | | | | | | | | | |
| PAIRWISE | all | .498 | .492 | .455 | .486 | .481 | .488 | .530 | .533 | .509 | .499 | .468 | .503 | .496 |
| | filter | .518 | .566 | .437 | .510 | .528 | .479 | .549 | .582 | .564 | .419 | .491 | .525 | .533 |
| LOGITS | $M_{50}$ | .541 | .568 | .524 | .505 | .570 | .531 | .482 | .464 | .566 | .491 | .419 | .577 | .532 |
| | $M_{10}$ | .559 | .669 | .432 | .528 | .571 | .649 | .496 | .427 | .400 | .444 | .310 | .630 | .582 |
| SCORE | $M_{50}$ | .621 | .552 | .493 | .554 | .643 | .502 | .574 | .580 | .604 | .514 | .626 | .424 | .546 |
| | $M_{10}$ | .654 | .551 | .458 | .579 | .701 | .485 | .659 | .524 | **.800** | **.632** | **.662** | .500 | .558 |
| **GPT-3.5 AS EVALUATOR** | | | | | | | | | | | | | | |
| PAIRWISE | all | .512 | .493 | .472 | .500 | .504 | .474 | .531 | .512 | .463 | .492 | .466 | .502 | .494 |
| | filter | .531 | .631 | .502 | .231 | .520 | .500 | .493 | .531 | .387 | .437 | .433 | .400 | .536 |
| LOGITS | $M_{50}$ | - | - | - | - | .548 | .511 | .570 | .475 | - | - | - | - | .507 |
| | $M_{10}$ | - | - | - | - | .541 | .538 | .570 | .430 | - | - | - | - | .505 |
| SCORE | $M_{50}$ | .424 | .566 | .473 | .552 | .578 | .522 | .608 | .600 | .264 | .502 | .503 | .551 | .547 |
| | $M_{10}$ | .585 | .632 | .517 | .555 | .583 | .550 | **.718** | .575 | .200 | .546 | **.662** | .573 | .590 |
| **GPT-4O AS EVALUATOR** | | | | | | | | | | | | | | |
| PAIRWISE | all | .605 | .593 | .507 | .551 | .646 | .512 | .515 | .577 | .434 | .501 | .526 | .537 | .562 |
| | filter | .691 | .689 | .533 | .602 | .712 | .536 | .558 | .661 | .417 | .490 | .604 | .549 | .624 |
| LOGITS | $M_{50}$ | - | - | - | - | - | - | - | - | .491 | .539 | .486 | .572 | .544 |
| | $M_{10}$ | - | - | - | - | - | - | - | - | .200 | .584 | .507 | .591 | .574 |
| SCORE | $M_{50}$ | .733 | .677 | **.544** | .605 | .793 | .591 | .617 | .661 | .547 | .520 | .581 | **.639** | .641 |
| | $M_{10}$ | **.793** | **.795** | .534 | .652 | **.835** | **.655** | .711 | .684 | .400 | .586 | .520 | .578 | **.709** |

Table 1: Accuracy of LLM-generated wrong-over-wrong preferences $Acc_{WoW}$. The three LLMs across on top are employed to generate answers (*generator*). Best results are in **bold**, second best are in underline, and incompatible or unavailable results are denoted as "-".[3] For pairwise comparison, margin "all" is without consistency checks and margin "filter" is with consistency checks. Overall accuracy is a weighted average across all datasets by the number of wrong-over-wrong pairs in each dataset. LLMs *do* have preliminary capabilities to distinguish varying shades of wrong with up to $Acc_{WoW} = 0.709$ achieved by the score-based approach, specifically $f_{GPT-4o}^{(s)}$ with $M_{10}$.

## 4 RESULTS

**Eliciting Wrong-over-Wrong Preferences** We present the accuracy of LLM-generated wrong-over-wrong preferences in Table 1.

- **Overall: Feasible to elicit wrong-over-wrong preference.** Most (approach, LLM) combinations yield wrong-over-wrong preferences that are significantly better than random guess: the average $Acc_{WoW}$ is 0.553 across datasets, and the best $Acc_{WoW}$ is 0.709 achieved by $f_{GPT-4o}^{(s)}$ with a margin of $M_{10}$. This suggests that LLMs, with the right approach, *do* possess preliminary capabilities to distinguish various shades of wrong.

- **Best eliciting method: Scored-based.** $f_{GPT-4o}^{(s)}$ with $M_{10}$ achieves the best $Acc_{WoW}$. The average $Acc_{WoW}$ across LLMs for consistency-based with $M_{10}$, logits-based with $M_{10}$, pairwise comparison with consistency checks, score-based with $M_{10}$, and heuristic are 0.554, 0.566, 0.564, 0.619, and 0.474, indicating that the score-based method outperforms all other approaches by at least 9.4% and provides clearly *non-random* judgments. This suggests that the token probabilities of aligned LLMs might not be well-calibrated (Sorensen et al., 2024; Feng et al., 2024) for wrong-over-wrong contexts, while LLM-as-a-judge methodologies (Li et al., 2024a) with score-based prompting offer the most promising solution.

---

[3]Answers for Bio Generation questions are open-ended and we cannot compute repetition; GPT-3.5 and GPT-4O are not open models for calculating logits and NLL.

| Method | KC | | | BG | | | COM$^2$ | | | NLG | | |
|---|---|---|---|---|---|---|---|---|---|---|---|---|
| | $p_{wrong}$↑ | Acc↑ | ECE↓ | $p_{wrong}$↑ | Acc↑ | ECE↓ | $p_{wrong}$↑ | Acc↑ | ECE↓ | $p_{wrong}$↑ | Acc↑ | ECE↓ |
| ORIGINAL | .466 | .555 | .235 | .532 | .027 | .576 | .312 | .669 | .053 | .750 | .142 | .649 |
| **SELF-GENERATOR** | | | | | | | | | | | | |
| PAIRWISE filter | .475 | **.627** | **.096** | .670 | .059 | .500 | .326 | .690 | .049 | .806 | .179 | .493 |
| SCORE $M_{50}$ | .529 | .597 | .251 | .661 | .043 | .580 | .325 | .660 | .039 | .800 | .203 | .551 |
| SCORE $M_{10}$ | .532 | .584 | .315 | .682 | .075 | .561 | **.357** | .681 | **.020** | .847 | .292 | .578 |
| ORACLE | .529 | .576 | .279 | **.695** | .108 | **.440** | .330 | .689 | .064 | .846 | .182 | .596 |
| **MIX-GENERATOR** | | | | | | | | | | | | |
| PAIRWISE filter | .533 | .574 | .201 | .634 | .075 | .535 | .355 | **.698** | .048 | .832 | .192 | .538 |
| SCORE $M_{50}$ | .528 | .590 | .175 | .619 | .065 | .523 | .329 | .669 | .067 | .827 | .221 | .585 |
| SCORE $M_{10}$ | .520 | .565 | .273 | .687 | **.129** | .560 | .346 | .677 | .065 | .843 | **.303** | .522 |
| ORACLE | **.537** | .581 | .185 | .691 | .086 | .472 | .328 | .697 | .067 | .832 | .226 | **.474** |

Table 2: Evaluation of wrong-over-wrong alignment on less wrong ($p_{wrong}$), more correct (Acc), and better calibration (ECE). The best results are in **bold**, second best are in underline, and green background indicates improvement over the original LLAMA3-8B. "Self-Generator" indicates that wrong-over-wrong pairs are generated from only LLAMA3-8B while "Mix-Generator" uses all 3 LLMs' answers. "Oracle" means aligning with proxy "ground-truth" wrong-over-wrong preference $\hat{f}$. Wrong-over-wrong alignment is helpful across the board, with up to 0.163, 0.161, and 0.175 improvement in reducing wrongness, increasing correct answers, and improving calibration.

- **Improve upon original eliciting method: consistency checks and score margins.** For the pairwise comparison method, we notice that LLMs are sensitive to response order and applying consistency checks by removing inconsistent judgments due to order-flipping improves 9.0% $\text{Acc}_{WoW}$ on average. Employing score margins to only consider the most separable cases also improves $\text{Acc}_{WoW}$ for logits-based, consistency-based, and score-based methods, with average improvements across datasets from $M_{50}$ to $M_{10}$ being 2.1%, 1.3%, and 6.9%, and from $M_{100}$ to $M_{10}$ being 5.8%, 72.4%, and 65.3%. The significant improvement from $M_{100}$ to $M_{10}$ is due to prevalent cases where the LLM gives the same score for wrong answers with different levels of wrongness. Appendix A also illustrates the trade-off effect of batch size.

- **Failed eliciting method: self-evaluation.** LLMs may not be good at producing accurate wrong-over-wrong preferences for their own generation. While GPT-4O provides good wrong-over-wrong preferences on generation from LLAMA3-8B and GPT-3.5, accuracy suffers for its own answers, with a 21.1% and 19.7% drop for score-based with $M_{10}$ and pairwise comparison with consistency checks on average across datasets. This echos findings that LLM self-critique and self-correction might not be satisfactory (Valmeekam et al., 2023; West et al., 2024; Huang et al., 2024a). More interestingly, employing weak LLMs to evaluate strong LLMs may be helpful (Khan et al., 2024), with LLAMA3-8B achieving 0.649 $\text{Acc}_{wow}$ on answers generated by GPT-4O.

**Alignment with Wrong-over-Wrong Preferences** We present the evaluation of wrong-over-wrong alignment in Table 2.

- **Overall: Wrong-over-wrong alignment helps to reduce wrongness, produce correct answers, and improve calibration.** After wrong-over-wrong alignment, LLAMA3-8B improves on average across datasets on $\Delta p_{wrong}$, $\Delta \text{Acc}$, and $-\Delta \text{ECE}$ by 0.074, 0.045, and 0.044. We also report the precision, recall and F1 improvement on Knowledge Crosswords and COM$^2$ datasets in Table 12. This validates that wrong-over-wrong alignment is moderately helpful across the three objectives. Even though we only align on wrong answers, we still magically end up making LLM generate 4.5% more correct answers on average. This finding highlights the potential of alignment with wrong answers generated by LLMs that come in large quantities.

- **Best preference data: Score-based achieves the least wrong and most correct answers while pairwise comparison achieves the best calibration.** Score-based preferences with $M_{10}$ on self-generated data achieve the best average improvement of $\Delta p_{wrong} = 0.090$ across datasets. $f_{GPT-4o}^{(s)}$ with $M_{10}$ on mix-generated data has the best average improvement of $\Delta \text{Acc} = 0.070$. This sug-

gests that the score-based method has the best $\text{Acc}_{WoW}$ and is most helpful in reducing wrongness and producing correct answers. However, $f_{GPT\text{-}4o}^{(p)}$ with consistency checks has the best average calibration improvement of $-\Delta\text{ECE} = 0.094$. We hypothesize that the margin used in score-based methods improves $\text{Acc}_{WoW}$ by avoiding cases with similar degrees of wrongness. This also indicates the models' tendency to avoid challenging distinctions, which can harm calibration.

- **Varied improvement across datasets: open-ended questions benefit more from wrong-over-wrong alignment compared to multiple-choice questions.** We observe substantial improvement on Bio Generation and NLGraph with up to 0.163, 0.161, and 0.175 for $\Delta p_{wrong}$, $\Delta\text{Acc}$, and $-\Delta\text{ECE}$. However, relatively less improvement is observed on Knowledge Crosswords and $\text{COM}^2$, with up to 0.071, 0.073, and 0.138 for $\Delta p_{wrong}$, $\Delta\text{Acc}$, and $-\Delta\text{ECE}$. This may be because Bio Generation and NLGraph are *open-ended* questions and there is more space for improvement compared to *multiple-choice* questions in Knowledge Crosswords and $\text{COM}^2$ where wrong options are limited and LLMs are confined to pre-defined options.

## 5 ANALYSIS

**Task Utility and Preference Accuracy** We explore the impact of two factors on the reliability of wrong-over-wrong preference: accuracy of the evaluator's generated answers (*task accuracy*) and the evaluator's confidence in its generated answers (*task confidence*). We visualize the correlations and present Pearson correlation coefficients (Sedgwick, 2012) in Figure 2. We observe a positive correlation between task accuracy and $\text{Acc}_{WoW}$, but a negative correlation between task confidence and $\text{Acc}_{WoW}$. This suggests that models that perform well on the task are also good at distinguishing various shades of wrong, while over-confident and under-calibrated models harm wrong-over-wrong preferences. We also find that LLM struggles to differentiate two answers with close wrongness levels in Appendix A.

**Preference Accuracy and Alignment Improvement** We examine how the quality of wrong-over-wrong judgements $\text{Acc}_{WoW}$ is related to improvements through wrong-over-wrong alignment. Figure 3 demonstrates a weak positive relationship between $\text{Acc}_{WoW}$ and improvement in $\Delta p_{wrong}$, and no significant correlation between $\text{Acc}_{WoW}$ and improvement in $\Delta\text{Acc}$ or $-\Delta\text{ECE}$. Surprisingly, this suggests that improvement resulting from wrong-over-wrong alignment is nuanced and is not sensitive to the absolute accuracy of wrong-over-wrong preference data. There is also a clear positive relationship between $\text{Acc}_{WoW}$ and $\Delta p_{wrong}$, $\Delta\text{Acc}$, and ECE on the Bio Generation dataset, indicating that sensitivity to wrong-over-wrong judgment quality varies by dataset.

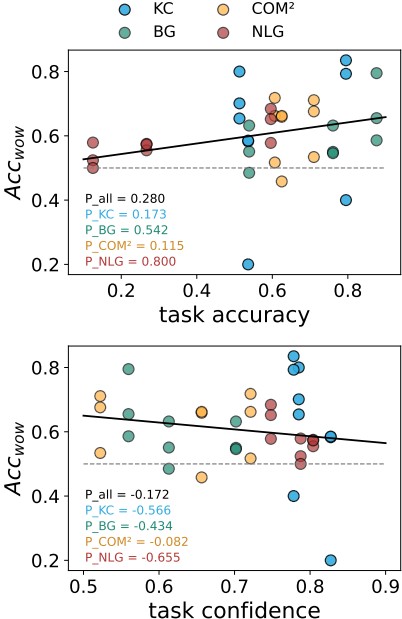

Figure 2: Correlation between task accuracy, confidence and $\text{Acc}_{WoW}$ of score-based eliciting with $M_{10}$. Data points are from all 3 LLMs we used to elicit wrong-over-wrong preferences. $P$ stands for Pearson correlation coefficient. The ability to elicit wrong-over-wrong preferences is positively correlated with task ability but negatively correlated with confidence.

**Right-over-Wrong Alignment** We conduct experiments on right-over-wrong preferences and a 50:50 mix of right-over-wrong and wrong-over-wrong preferences. The results in Table 3 reveal that: (1) Right-over-wrong alignment has the best average improvement in more correct $\Delta p_{wrong} = 0.090$ on mix-generated data, while on self-generated data, wrong-over-wrong alignment yields the best $\Delta p_{wrong} = 0.068$. Mixing generators is especially helpful to right-over-wrong alignment, with average $\Delta\text{Acc}$ improved by 55.7%. (2) Right-over-wrong alignment has the best average improvement in more correct $\Delta\text{Acc} = 0.134$, while a mixture of right-over-wrong and wrong-over-wrong preferences achieves the best average improvement in calibration $-\Delta\text{ECE} = 0.107$. This partly resolves the previous finding that alignment hurts calibration (Sorensen et al., 2024; Feng et al.,

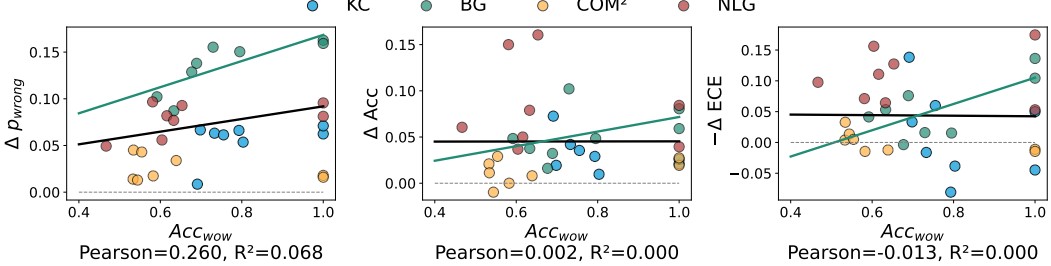

Figure 3: Correlation between $\text{Acc}_{WoW}$ and improvement after wrong-over-wrong alignment in less wrong $\Delta p_{wrong}$, more correct $\Delta \text{Acc}$, and better calibration $-\Delta \text{ECE}$. Data points are sourced from all 4 methods ($f_{GPT\text{-}4o}^{(p)}$ with consistency checks, $f_{GPT\text{-}4o}^{(s)}$ with $M_{50}$ and $M_{10}$, and oracle $\hat{f}$), and the oracle method is considered as $\text{Acc}_{WoW} = 1.0$. The black line is the linear regression on all four datasets while the green line is the linear regression on Bio Generation dataset. Wrong-over-wrong alignment is not sensitive to the accuracy of wrong-over-wrong preference.

2024): we hypothesize that this is due to missing wrong-over-wrong preferences in alignment data that make LLMs unable to disentangle varying shades of wrong for fine-grained calibration.

**Generalization to Unseen Data** We examine how wrong-over-wrong alignment can generalize to other unseen tasks in the same domain. We employ Hellaswag (Zellers et al., 2019) and the maximum flow subtask in NL-Graph (Wang et al., 2023a) as two unseen datasets while being in the same domain as the datasets employed for alignment tuning, COM$^2$ and shortest path. Results in Table 5 demonstrate that wrong-over-wrong alignment could indeed generalize to unseen data, with an average improvement of 0.118, 0.002, and 0.029 on $\Delta p_{wrong}$, $\Delta \text{Acc}$, and $-\Delta \text{ECE}$.

| Method | KC | | | COM$^2$ | | | NLG | | |
|---|---|---|---|---|---|---|---|---|---|
| | $p_{wrong}\uparrow$ | Acc$\uparrow$ | ECE$\downarrow$ | $p_{wrong}\uparrow$ | Acc$\uparrow$ | ECE$\downarrow$ | $p_{wrong}\uparrow$ | Acc$\uparrow$ | ECE$\downarrow$ |
| ORIGINAL | .466 | .555 | .235 | .312 | .669 | .053 | .750 | .142 | .649 |
| BEST W | .529 | .597 | .251 | .355 | .698 | .048 | .847 | .292 | .578 |
| **SELF-GENERATOR** | | | | | | | | | |
| R+W (PAIRWISE) | .493 | .706 | .065 | .359 | .705 | .057 | .806 | .179 | .493 |
| R+W (SCORE $M_{50}$) | .540 | .690 | .055 | .335 | .673 | .037 | .814 | .139 | .667 |
| R+W (SCORE $M_{10}$) | .503 | .665 | .174 | .340 | .687 | .070 | .815 | .137 | .650 |
| R+W (ORACLE) | **.607** | .785 | .060 | .357 | .703 | .055 | .760 | .158 | .549 |
| R | .579 | **.805** | .079 | .373 | .692 | **.029** | .777 | .153 | .652 |
| **MIX-GENERATOR** | | | | | | | | | |
| R+W (PAIRWISE) | .530 | .705 | .053 | .327 | .658 | .091 | .825 | .218 | .493 |
| R+W (SCORE $M_{50}$) | .536 | .705 | .094 | .326 | .716 | .070 | .836 | .263 | .548 |
| R+W (SCORE $M_{10}$) | .559 | .711 | .104 | .343 | .655 | .034 | .842 | **.308** | .496 |
| R+W (ORACLE) | .567 | .740 | **.025** | .344 | **.748** | .113 | .826 | .274 | **.490** |
| R | .568 | .787 | .110 | **.374** | .711 | .183 | **.856** | .268 | .537 |

Table 3: Evaluation of right-over-wrong alignment on less wrong ($p_{wrong}$), more correct (Acc), and better calibration (ECE). "w" and "r" stand for wrong-over-wrong and right-over-wrong data. Wrong-over-wrong alignment is a good supplement to existing right-over-wrong alignment and best employed in conjunction.

**Qualitative Examples** We present qualitative examples in Table 4, showing that WoW-aligned models could generate less wrong, more correct, and better calibrated answers. More successful and failed examples can be found in Appendix A.

## 6 RELATED WORK

**LLM Alignment** In recent years, aligning LLMs with human preferences has become an important research question, driven by the need for safety (Dai et al., 2024; Qi et al., 2024; Huang et al., 2024b), honesty (Yang et al., 2023; Wen et al., 2024), factuality (Liang et al., 2024; Lin et al., 2024), diversity (Ding et al., 2024), etc. The initial methods, such as Reinforcement Learning from Human Feedback (RLHF) (Ziegler et al., 2019; Stiennon et al., 2020; Ouyang et al., 2022; Munos et al., 2024; Chakraborty et al., 2024), leverage human preferences to train reward models and employ algorithms such as Proximal Policy Optimization (PPO) (Schulman et al., 2017) for alignment. Recent alignment methods move away from explicit reward models, as seen in Direct Preference Optimization (DPO) (Rafailov et al., 2023). Alongside this, Reinforcement Learning from AI Feedback (RLAIF) (Bai et al., 2022; Lee et al., 2023) and LLM-as-a-Judge (Dubois et al., 2023; Bai et al., 2023) frameworks have introduced AI-driven feedback systems, allowing LLMs to give feedback and improve themselves with minimal human intervention. Techniques such as Self-Alignment (Li et al., 2024b; Wang et al., 2024c; Yuan et al., 2024; Sun et al., 2024a), which involves models evaluating and aligning with their own outputs, represent a further step towards scalable oversight.

| Question | In an undirected graph, the nodes are numbered from 0 to 4, and the edges are represented by (start node, end node, weight): (0, 2, 2), (0, 5, 4), (0, 3, 1), (0, 1, 3), (1, 2, 4), (1, 5, 1), (1, 4, 4), (2, 4, 3), (2, 3, 1), (3, 5, 2), (3, 4, 3), (4, 5, 2). Q: Give the shortest path from node 0 to node 1. Please also give the total weight of the shortest path. [Correct answer: 4 $(0 \rightarrow 3 \rightarrow 4)$.] |
|---|---|
| **Original Answer** | [CoT steps]. The final answer is: shortest path: $0 \rightarrow 2 \rightarrow 3 \rightarrow 4$; total weight: 6. |
| **Less Wrong** | [CoT steps]. Therefore, the answer is: The shortest path from node 0 to node 4 is $0 \rightarrow 3 \rightarrow 2 \rightarrow 4$, with a total weight of 5. [The path found is shorter: $6 \rightarrow 5$.] |
| **More Correct** | [CoT steps]. Final answer: The shortest path from node 0 to node 4 is Node $0 \rightarrow$ Node $3 \rightarrow$ Node 4, with a total weight of 4. [Successfully find the shortest path.] |
| **Better Calibrated** | [CoT steps]. The total weight of the shortest path is: $0 \rightarrow 2$ (weight 2), $2 \rightarrow 3$ (weight 1), $3 \rightarrow 4$ (weight 3). Total weight: $2 + 1 + 3 = 6$. [Even though the answer is equally wrong compared to the original answer, NLL increases from $0.255 \rightarrow 0.373$.] |

Table 4: Qualitative examples on NLGraph where WoW-aligned models become less wrong, more correct, and better calibrated on NLGraph. We provide explanations in [red].

Instead of alignment objective, we investigate an orthogonal research question to the improvement in alignment approaches and focus on the preference data: in challenging tasks and contexts when correct and high-quality answers are not readily available (Baheti et al., 2024; Agarwal et al., 2024), could LLMs distinguish various shades of wrong and align with wrong-over-wrong preferences? Prior work has also established the utility and representativeness of fine-grained and continuous preferences over categorical labels in uncertain (Lin et al., 2022; Tian et al., 2023) and subjective contexts (Sorensen et al., 2024; Feng et al., 2024).

**Easy-to-Hard Generalization** Most of these advancements in alignment benefit from the availability of "correct" answers. In contrast, our work posits that "correct" answers are inevitably unavailable or unreliable for emerging and challenging tasks. Bowman et al. (2022) points out advanced models may engage in tasks that are beyond current evaluation capabilities. Easy-to-hard generalization (Burns et al., 2024; Hase et al., 2024; Xu et al., 2024b; Sun et al., 2024b) introduces a series of solutions hoping to improve LLMs on hard tasks by finetuning on easy in-domain tasks. Our work uniquely points out a new path for training future LLMs on hard tasks when correct answers are unavailable, arguing for wrong-over-wrong alignment based on synthetic preference data as a complementary training objective.

| Method | HellaSwag | | | Maximum Flow | | |
|---|---|---|---|---|---|---|
| | $p_{wrong}\uparrow$ | Acc$\uparrow$ | ECE$\downarrow$ | $p_{wrong}\uparrow$ | Acc$\uparrow$ | ECE$\downarrow$ |
| ORIGINAL | .230 | .737 | .089 | .112 | .069 | .663 |
| **SELF-GENERATOR** | | | | | | |
| PAIRWISE | .243 | .729 | .098 | .344 | .066 | **.567** |
| SCORE $M_{50}$ | .220 | .679 | .047 | .330 | .083 | .637 |
| SCORE $M_{10}$ | .264 | .719 | .068 | .342 | **.109** | .673 |
| ORACLE | .227 | .729 | .023 | .151 | .049 | .659 |
| **MIX-GENERATOR** | | | | | | |
| PAIRWISE | .247 | .706 | .082 | **.348** | .094 | .533 |
| SCORE $M_{50}$ | .204 | .729 | .090 | .346 | .083 | .621 |
| SCORE $M_{10}$ | .250 | **.771** | .117 | .326 | .089 | .627 |
| ORACLE | **.267** | .753 | .099 | .202 | .074 | .582 |

Table 5: Generalization to unseen datasets, from $COM^2$ to HellaSwag and from shortest path to maximum flow. While tuned only on $COM^2$ and shortest path, the aligned models improve on the two unseen datasets as well, with an average improvement of 0.118, 0.002, and 0.029 on $\Delta p_{wrong}$, $\Delta$Acc, and $-\Delta$ECE.

## 7 CONCLUSION

With the growing race towards bigger, better LLMs capable of solving a wider range of tasks, it becomes evident that the availability of carefully curated data is a major bottleneck. Our work investigates the potential to alleviate this limitation by eliciting preferences among wrong answers with an LLM and aligning with these wrong-over-wrong preferences. We empirically investigate wrong-over-wrong alignment with seven LLMs and eight datasets. We find that LLMs *do* have preliminary capability to rank wrong answers and produce reliable wrong-over-wrong preferences. The strongest approach such as score-based LLM-as-a-judge achieves up to 70.9% accuracy across datasets. In addition, alignment with such wrong-over-wrong preferences is helpful in calibrating the models, e.g., reducing answer "wrongness" by 9.0%, producing 7.0% more correct answers, and improving overall model calibration by 9.4% on average. We envision future wrong-over-wrong alignment methods as an important addition to advance LLM capabilities on challenging tasks where correct answers or rich annotations are not readily available.

## LIMITATIONS AND ETHICS STATEMENT

**Imperfect proxies.**  The proxy functions we use to measure correctness are inherently imperfect. While they serve as reasonable approximations for distinguishing varying degrees of wrongness, they are not definitive, e.g. heuristic-based proxies may merely focus on the final answer and overlook the reasoning steps and model-based proxies are inaccurate and may suffer from domain shift problem. As a result, some wrong-over-wrong preferences may be misjudged, potentially affecting the evaluation of $\text{Acc}_{WoW}$ and analysis results.

**Sensitivity to hyperparameters.**  The alignment process we employ is sensitive to hyperparameter selection. This dependence can lead to variability in results across different models and datasets, making it challenging to guarantee consistent performance improvements. Further exploration of more robust hyperparameter configurations (Falkner et al., 2018; Arango et al., 2024) is necessary.

**Limited scope of experimental datasets.**  Experiments are conducted on datasets where ground-truth answers exist, and proxy functions can be employed to approximate wrongness. However, this setup doesn't fully reflect the application scenarios we aim to address — tasks where no clear ground-truth answers are available (Wang et al., 2024a), or the problems are so challenging that even expert human annotators or LLMs might struggle (Welleck et al., 2021). Evaluating the generalizability of our framework in these conditions remains an important avenue for future work.

**Unique focus on knowledge and reasoning problems.**  We focus on tasks and problems with absolute and indisputable correct answers in this work, such as multi-hop QA (Ding et al., 2023) and graph reasoning (Wang et al., 2023a). Consequently, existing alignment works in these domains often assume access to correct answers and construct right-over-wrong pairs for preference learning (Wang et al., 2023c; Cheng et al., 2024; Lin et al., 2024). However, objective "correct answers" are often not feasible in general instruction following tasks, thus they employ human preference data between a pair of responses that is not necessarily right-over-wrong. Our scope is to investigate wrong-over-wrong alignment specifically focusing on the first type of knowledge/reasoning problems with absolute correctness while we leave general instruction following as future work.

**Incomparable wrongness in social contexts.**  In social contexts, wrongness is often subjective and potentially incomparable, which introduces biases and fairness concerns in wrong-over-wrong alignment. For instance, when judging the wrongness of statements related to sensitive topics such as politics or identity, cultural biases may influence the model's preferences (Xu et al., 2021; Bender et al., 2021; Feng et al., 2023). A statement like "People from *certain* neighborhoods are more likely to commit crimes" might be judged less wrong than "All people from *certain* neighborhoods are criminals" but these evaluations reflect specific perspectives that may not be universally shared or appropriate in all contexts. Furthermore, aligning models with such judgments risks reinforcing harmful stereotypes or systemic biases. Ensuring fairness and transparency in these judgments is critical, and models should be designed to recognize when wrongness is subjective and abstain from making harmful comparisons.

## REPRODUCIBILITY STATEMENT

We provide all necessary details for the implementation and evaluation of our proposed wrong-over-wrong alignment approach in Appendix B. Specifically, Appendix B includes information of 8 datasets splits and preprocessing steps, 7 LLM checkpoints, hyperparameters selection and detailed configuration for each table. Moreover, the prompts for preference elicitation methods are explained in Table 13 and Table 14. Our codes, including scripts for dataset generation, preference eliciting, preference accuracy evaluation, alignment, and evaluation of alignment are available publicly in repo https://github.com/yaojh18/Varying-Shades-of-Wrong to facilitate easy verification of our results.

## ACKNOWLEDGMENTS

This research was developed with funding from the Defense Advanced Research Projects Agency's (DARPA) SciFy program (Agreement No. HR00112520300). The views expressed are those of the

author and do not reflect the official policy or position of the Department of Defense or the U.S. Government. We also gratefully acknowledge gift funding from Google, OpenAI, Microsoft Research, and the Allen Institute for AI (Ai2).

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

# A   ANALYSIS (CONT.)

**More general LLMs as evaluators**   We present the rest of elicited wrong-over-wrong preferences, including evaluators being GEMINI-FLASH, GEMINI-PRO, and MISTRAL-7B. The results in Table 9 demonstrate that GPT-4O is still the best LLM at evaluating wrong-over-wrong preference and surpasses other LLMs by at least 12.0% on overall $\text{Acc}_{WoW}$. We can also see leveraging GEMINI-FLASH, GEMINI-PRO to evaluate answers generated by GPT-4O successfully mitigate the poor accuracy of self-evaluation.

**Domain-specific LLMs as evaluators**   To better figure out how $\text{Acc}_{wow}$ is relate to evaluators, we instruction finetune GEMMA-7B on 7 different datasets (Ivison et al., 2023), namely Code-Alpaca (Chaudhary, 2023), FLAN (Chung et al., 2024), Gemini-Alpaca, LIMA (Zhou et al., 2023), Open Assistant 1 (Köpf et al., 2023), Science literature and WizardLM Evol-Instruct V2 (Xu et al., 2023). Then we elicit wrong-over-wrong preference from them on Knowledge Crosswords, Bio Generation, $\text{COM}^2$ and NLGraph datasets with GPT-3.5 being generator. We only experiment on score-based eliciting. Results in Table 6 demonstrate that domain-specific LLMs as evaluators show considerable variance in their wrong-over-wrong preference elicitation accuracy across datasets. For example, GEMMA-7B fintuned on FLAN achieves the best wrong-over-wrong accuracy on Bio Generation and NLGraph while unable to follow instructions on Knowledge Crosswords and $\text{COM}^2$. This suggests finetuning as a way to improve LLM-based wrong-over-wrong accuracy in certain domains.

| Margin | KC | BG | $\text{COM}^2$ | NLG | overall |
|--------|----|----|------|------|---------|
| *Code-Alpaca* | | | | | |
| $M_{100}$ | .183 | .258 | .221 | .174 | .220 |
| $M_{50}$ | .367 | .516 | .442 | .348 | .439 |
| $M_{10}$ | .425 | .544 | .466 | .482 | .505 |
| *FLAN* | | | | | |
| $M_{100}$ | .000 | .098 | .000 | .096 | .080 |
| $M_{50}$ | .000 | .197 | .000 | .193 | .160 |
| $M_{10}$ | .000 | **.585** | .000 | **.610** | .487 |
| *Gemini-Alpaca* | | | | | |
| $M_{100}$ | .297 | .356 | .396 | .347 | .349 |
| $M_{50}$ | .512 | .492 | .555 | .527 | .510 |
| $M_{10}$ | .534 | .458 | **.679** | .575 | .519 |
| *LIMA* | | | | | |
| $M_{100}$ | .110 | .079 | .215 | .099 | .098 |
| $M_{50}$ | .221 | .159 | .428 | .198 | .197 |
| $M_{10}$ | **.610** | .508 | .343 | .486 | .501 |
| *Open Assist* | | | | | |
| $M_{100}$ | .240 | .175 | .220 | .180 | .187 |
| $M_{50}$ | .481 | .351 | .441 | .360 | .375 |
| $M_{10}$ | .500 | .563 | .602 | .553 | **.555** |
| *Science* | | | | | |
| $M_{100}$ | .241 | .236 | .557 | .248 | .262 |
| $M_{50}$ | .482 | .472 | .561 | .482 | .482 |
| $M_{10}$ | .515 | .557 | .574 | .470 | .525 |
| *WizardLM* | | | | | |
| $M_{100}$ | .194 | .278 | .322 | .239 | .259 |
| $M_{50}$ | .388 | .508 | .474 | .478 | .482 |
| $M_{10}$ | .579 | .551 | .523 | .479 | .529 |

Table 6: Accuracy of wrong-over-wrong preference elicited from GEMMA-7B fintuened on different instruction datasets. Finetuned LLMs have very different wrong-over-wrong preference accuracy across domains.

**More challenging datasets**   We investigate LLMs' capability to give wrong-over-wrong preferences on more challenging datasets, NLGraph (full) and ChessPuzzle (CP). NLGraph (full) employs all min/max questions in NLGraph, including very challenging questions like "finding the maximum flow between two nodes in a weighted undirected graph" and "finding an assignment of jobs to applicants such that the maximum number of applicants find the job they are interested in". Chess Puzzle contains chess puzzles from the lichess.org website (Lichess Team, 2023), where the inputs are board state represented in Forsyth–Edwards Notation (FEN) and LLMs are asked to generate the optimal next move for the current player. However, we find that LLMs tend to give invalid moves, which makes it hard to evaluate wrongness. Thus we employ Stockfish (Romstad et al., 2021), a SOTA chess engine, to generate the 4 best moves for the current player and ask an LLM to choice from given options. Evaluators' task accuracy is 0.204 (random guess 0.000) on NLGraph (full) and 0.461 (random guess 0.250) on Chess Puzzle. The correctness proxy for NLGraph (full) is $p^{(nl)}(\boldsymbol{a} \mid \boldsymbol{q}) = -|v_{\boldsymbol{a}} - v_{gt}|$, where $v_{\boldsymbol{a}}$ is the extracted final answer and $v_{gt}$ is the ground-truth answer. The correctness proxy for Chess Puzzle is $p^{(nl)}(\boldsymbol{a} \mid \boldsymbol{q}) = wr_{Stockfish}(\boldsymbol{a} \mid \boldsymbol{q})$, where $wr_{Stockfish}$ is the win rate predicted by Stockfish. Experiment results in Table 10 demonstrate that the evaluator's

$Acc_{WoW}$ drops significant on challenging tasks, e.g. $f_{GPT\text{-}4o}^{(s)}$ with $M_{10}$ drops by 23.7%. The average $Acc_{WoW}$ across datasets for pairwise comparison with consistency check, logits-based with $M_{10}$ and scored-based $M_{10}$ are 0.585, 0.546, 0.550, indicating pairwise comparison may be more suitable for challenging tasks.

**Batch size for score-based eliciting** We also investigate the effect of batch size $b$ on score-based preference eliciting. We employ LLAMA3-8B as generator and GPT-4O as evaluator. For simplicity, we only experiment on Knowledge Crosswords and NLGraph (shortest path) dataset. Experiment results in Table 7 indicate that 5 is the best batch size for our experiment setup, which verifies our assumption that batch size has a trade-off effect on scored-based eliciting.

**Consistency of pairwise comparison** Pairwise comparison eliciting displays high sensitivity to positional bias. We report the proportions of consistent wrong-over-wrong preference after flipping for LLAMA3-8B, GPT-3.5 and GPT-4O are 0.286, 0.128, 0.598. Stronger LLMs can generate more consistent preference, and small LLMs are not reliable evaluators.

**Wrongness margin affecting $Acc_{WoW}$** The accuracy of wrong-over-wrong judgement can also be affected by "wrongness margin" of a wrong-over-wrong pair. For example, a close-to-correct answer (e.g. "the sun almost all rises in the east") is obviously better than an absolutely wrong answer (e.g. "the sun rises in the west"), while two wrong answers with close wrongness (e.g. "the sun sometimes rises in the east" and "the sun sometimes perhaps rises in the east") are hard to differentiate. We experiment on Knowledge Crosswords dataset and evaluate the wrongness margin is defined as the the absolute difference of correctly filled blanks. The results in Table 11 demonstrate that a larger wrongness margin means easier evaluation, with average $Acc_{WoW}$ = 0.696, 0.636 and 0.518 for wrongness margin being 3, 2, 1. This trend is particularly evident for highly wrong answer comparison, with $3 \succ 2$ having an accuracy of 0.533 while $1 \succ 0$ only has an accuracy of 0.463.

| Margin | KC | NLG | Overall |
|---|---|---|---|
| *batch size = 1* | | | |
| $M_{100}$ | 0.447 | 0.318 | 0.383 |
| $M_{50}$ | 0.643 | 0.596 | 0.620 |
| $M_{10}$ | 0.719 | **0.701** | 0.710 |
| *batch size = 2* | | | |
| $M_{100}$ | 0.482 | 0.364 | 0.423 |
| $M_{50}$ | 0.654 | 0.575 | 0.615 |
| $M_{10}$ | 0.708 | 0.623 | 0.666 |
| *batch size = 5* | | | |
| $M_{100}$ | 0.479 | 0.383 | 0.431 |
| $M_{50}$ | 0.733 | 0.605 | 0.669 |
| $M_{10}$ | 0.793 | 0.652 | **0.723** |
| *batch size = 10* | | | |
| $M_{100}$ | 0.404 | 0.299 | 0.352 |
| $M_{50}$ | 0.723 | 0.598 | 0.661 |
| $M_{10}$ | **0.804** | 0.634 | 0.719 |

Table 7: Accuracy of scored-based wrong-over-wrong preference on Knowledge Crosswords and NLGraph (shortest path) dataset. The generator is LLAMA3-8B and the evaluator is GPT-4O. The number of answers generated to each question is 10. $b = 5$ yields the best overall $Acc_{WoW}$.

**No or negative effect of mixing generators is observed.** Experiments in Table 2 demonstrate no effect from mixing generators for $f_{GPT\text{-}4o}^{(p)}$ with consistency checks, $f_{GPT\text{-}4o}^{(s)}$ with $M_{10}$ and oracle, with little change in average proxy scores discovered. And Table 2 also indicates an negative effect on $f_{GPT\text{-}4o}^{(s)}$ with $M_{50}$ with average improvement on $\Delta Acc$ across datasets drops by 28.9%. This suggests *learning from the model's own mistakes is perhaps more important for wrong-over-wrong alignment.*

**Preference Optimization Approach** To investigate the influence of different preference optimization methods, we also experiment wrong-over-wrong alignment with IPO (Azar et al., 2024),

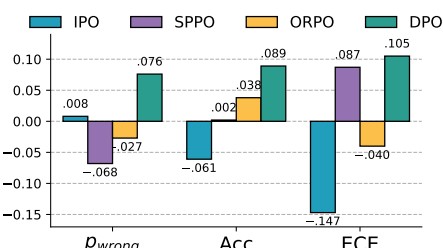

Figure 4: Evaluation different of preference optimization methods on less wrong, more correct and better calibration. The number is averaged over six experiment setups ((pairwise comparison with consistency check, score-based with $M_{10}$, score-based with $M_{50}$) $\times$ (Self-Generator, Mix-Generator)) on NLGraph dataset.

ORPO (Hong et al., 2024) SimPO (Meng et al., 2024) and SPPO (Wu et al., 2024) on the NL-Graph dataset. Results in Figure 4 demonstrate that preference optimization methods other than DPO are sensitive to noise and could not be employed for wrong-over-wrong alignment without hyperparameter tuning. SimPO collapses on all 6 experiment setups, IPO collapses on 4 of them, and SPPO and ORPO show negative improvement compared to original LLAMA3-8B. This suggests wrong-over-wrong alignment could be noisy and we should adopt stable preference optimization methods like DPO (Xu et al., 2024a; Chowdhury et al., 2024).

**Alignment on domains without clear wrongness distinctions** We wonder if the effectiveness of wrong-over-wrong alignment is universal and doesn't rely on clear wrongness distinctions within the task domain. To better prove this, we implement wrong-over-wrong alignment on SciBench (Wang et al., 2024b) and MedMCQA (Pal et al., 2022) where no such proxy to evaluate the wrongness of answers exists. SciBench is a college-level scientific benchmark emphasizing scientific theorem application and numerical computation, while MedMCQA is a multiple-choice QA dataset in the medical domain. Experiment results in Table 8 indicate that wrong-over-wrong alignment is not universally helpful in domains without clear right-wrong distinctions. This suggests we should investigate more fine-grained separability among wrong answers and design a more sophisticated LLM-as-a-judge prompt.

| Method | MedMCQA | | SciBench | |
|---|---|---|---|---|
| | Acc$_\uparrow$ | ECE$_\downarrow$ | Acc$_\uparrow$ | ECE$_\downarrow$ |
| ORIGINAL | 0.566 | 0.083 | 0.105 | 0.634 |
| **SELF-GENERATOR** | | | | |
| SCORE $M_{50}$ | **0.565** | 0.111 | 0.095 | 0.607 |
| SCORE $M_{10}$ | 0.550 | **0.082** | 0.068 | 0.660 |
| **MIX-GENERATOR** | | | | |
| SCORE $M_{50}$ | 0.548 | 0.083 | **0.108** | **0.591** |
| SCORE $M_{10}$ | 0.553 | 0.098 | 0.095 | 0.688 |

Table 8: Generalization datasets without distinctive wrong-over-wrong separability. Wrong-over-wrong alignment is not helpful for domains without clear right-wrong distinctions.

**Qualitative examples** We present qualitative examples demonstrating the improvement of wrong-over-wrong alignment on Knowledge Crosswords, Bio Generation, COM$^2$, NLGraph from Table 15 to Table 25.

# B EXPERIMENT DETAILS

**Model Details** We employ 7 LLMs in the experiments, the detailed information of LLMs we used in this paper is as follows: 1) LLAMA3-8B, through the META-LLAMA/META-LLAMA-3-8B checkpoint on Huggingface (Wolf et al., 2020); 2) MISTRAL-7B, through the MISTRALAI/MISTRAL-7B-INSTRUCT-V0.3 checkpoint on Huggingface; 3) GEMMA-7B, we directly employ 7 instruction finetuned GEMMA-7Bs from Ivison et al. (2023); 4) GPT-3.5, through the GPT-3.5-TURBO-0125 checkpoint on OpenAI API; 5) GPT-4O, through the GPT-4O-2024-05-13 checkpoint on OpenAI API; 6) GEMINI-FLASH through the GEMINI-1.5-FLASH-001 by the Vertex AI API; 7) GEMINI-PRO through the GEMINI-1.5-PRO-001 by the Vertex AI API.

**Dataset Details** We employ 8 datasets in the experiments, 4 for the main experiments and 4 left in the appendix. The detailed information is:

- **Knowledge Crosswords (KC)** (Ding et al., 2023): We downsample questions containing exactly three blanks, and we create four answer options by randomly choosing 0, 1, 2, or 3 correctly filled blanks. We sample 625 questions from the official dataset and split them into train, validation, test sets with an 8:1:1 ratio.

- **NLGraph (NLG)** (Wang et al., 2023a; Zhang et al., 2024) We employ the "shortest path" subset for main experiments denoted as NLGraph (shortest path) and "shortest path", "maximum flow" and "matching" subsets for experiments in the appendix denoted as NLGraph (full). Correctness proxy for "maximum flow" and "matching" subset is: $p^{(mf)}(\boldsymbol{a} \mid \boldsymbol{q}) = 1 - \frac{|f_{gt} - f_{\boldsymbol{a}}|}{f_{gt}}$, with $f_{\boldsymbol{a}}$ being value given in $\boldsymbol{a}$ and $f_{gt}$ being the ground-truth. We sample 380, 350 and 600 questions in "shortest path", "maximum flow" and "matching" subsets and split them into train, validation, and test sets with an 8:1:1 ratio.

- **Bio Generation (BG)** We employ the bio generation dataset and retrieved documents provided in FActScore (Min et al., 2023). We sample 625 questions from the official dataset and split them into train, validation, test sets with an 8:1:1 ratio.

- **COM$^2$** (Fang et al., 2024): We employ the "2i" and "3i" question subsets of COM$^2$, where we empirically find that Vera could generate more accurate scores. We sample 625 questions from the official dataset and split them into train, validation, test sets with an 8:1:1 ratio.

- **Hellaswag** (Zellers et al., 2019): Correctness proxy function for HellaSwag is: $p^{((hs))}(\boldsymbol{a} \mid \boldsymbol{q}) = \text{Vera}(\boldsymbol{a} \mid \boldsymbol{q}) \in [0, 1]$. We sample 125 questions from the official validation split and split them into validation, test sets with a 1:1 ratio.

- **Chess Puzzle** (Lichess Team, 2023): We downsample the beginner-level chess puzzles with ELO ranging from 600 to 1000. We sample 625 questions from the official dataset and split them into train, validation, test sets with an 8:1:1 ratio.

- **SciBench** (Wang et al., 2024b): We sample 625 questions from the official dataset and split them into train, validation, and test sets with an 8:1:1 ratio.

- **MedMCQA** (Pal et al., 2022): We sample 500 questions from the official train split as train set. We sample 125 questions from the official validation split into validation, test sets with a 1:1 ratio.

**Implementation Details**

- **Pairwise comparison**. The exact prompt we use for pairwise comparison is in Table 13.

- **Score-based** The exact prompt we use for score-based eliciting is in Table 14.

- **Generation of wrong-over-wrong dataset** The pipeline used to generate $\mathcal{D}_{WoW}$ is described in Algorithm 1.

- **QLora finetuning** We employ Unsloth and Transformers libraries for preference optimization. We apply grid search on learning rate (1e-4, 5e-5, 1e-5), learning rate scheduler (cosine, cosine with restart and reduce lr on plateau), weight decay (0, 1e-5, 1e-3) and number of train epochs (1, 3, 5) for main experiments and right-over-wrong alignment experiments. We use random seed = 42 for all of our experiments.

**Experiment Details** In Table 1, we use batch size = 5 for all score methods due to optimal empirical results. In Table 2, we define FActScore > 0.9 as "correct" on Bio Generation dataset; all the preferences are elicited from GPT-4O. In Table 3 the best wrong-over-wrong alignment setup for Knowledge Crosswords, Bio Generation, COM$^2$ and NLGraph are score-based with $M_{50}$ on self-generated data, pairwise comparison with consistency check on mix-generated data and score-based with $M_{10}$ on self-generated data; Bio Generation dataset is omitted because there is not well-defined ground-truth on this dataset. Correlations between task accuracy, confidence and other wrong-over-wrong preference eliciting methods are displayed in Figure 5 and Figure 6.

| Method | Margin | LLAMA3-8B | | | | GPT-3.5 | | | | GPT-4O | | | | Overall |
|---|---|---|---|---|---|---|---|---|---|---|---|---|---|---|
| | | KC | BG | COM² | NLG | KC | BG | COM² | NLG | KC | BG | COM² | NLG | |
| **MODEL-INDEPENDENT** | | | | | | | | | | | | | | |
| HEURISTIC | $M_{100}$ | .491 | .486 | .488 | .498 | .519 | .504 | .473 | .466 | .547 | .471 | .473 | .502 | .489 |
| CONSISTENCY | $M_{100}$ | .296 | - | .287 | .360 | .318 | - | .279 | .336 | .311 | - | .325 | .313 | .328 |
| **LLAMA3-8B** *as Evaluator* | | | | | | | | | | | | | | |
| LOGITS | $M_{100}$ | .527 | .540 | .525 | .508 | .546 | .516 | .506 | .477 | .528 | .491 | .440 | .546 | .519 |
| SCORE | $M_{100}$ | .479 | .386 | .385 | .415 | .496 | .376 | .48 | .434 | .387 | .373 | .525 | .273 | .400 |
| **GPT-3.5** *as Evaluator* | | | | | | | | | | | | | | |
| LOGITS | $M_{100}$ | - | - | - | - | .528 | .508 | .552 | .479 | - | - | - | - | .504 |
| SCORE | $M_{100}$ | .212 | .352 | .287 | .323 | .289 | .327 | .458 | .422 | .132 | .273 | .269 | .307 | .335 |
| **GPT-4O** *as Evaluator* | | | | | | | | | | | | | | |
| LOGITS | $M_{100}$ | - | - | - | - | - | - | - | - | .472 | .526 | .478 | .540 | .526 |
| PAIRWISE | all | .605 | .593 | .507 | .551 | .646 | .512 | .515 | .577 | .434 | .501 | .526 | .537 | .562 |
| | filter | .691 | .689 | .533 | .602 | .712 | .536 | .558 | .661 | .417 | .490 | .604 | .549 | .624 |
| SCORE | $M_{100}$ | .479 | .385 | .450 | .383 | .540 | .351 | .484 | .408 | .481 | .353 | .467 | .369 | .393 |
| | $M_{50}$ | .733 | .677 | .544 | .605 | .793 | .591 | .617 | .661 | .547 | .520 | .581 | **.639** | .641 |
| | $M_{10}$ | **.793** | **.795** | .534 | **.652** | **.835** | **.655** | .711 | **.684** | .400 | .586 | .520 | .578 | **.709** |
| **MISTRAL-7B** *as Evaluator* | | | | | | | | | | | | | | |
| PAIRWISE | all | .463 | .471 | .462 | .492 | .447 | .482 | .524 | .501 | .453 | .503 | .471 | .514 | .482 |
| | filter | .551 | .427 | .451 | .505 | .531 | .491 | .574 | .538 | .542 | .365 | .542 | .560 | .514 |
| SCORE | $M_{100}$ | .351 | .356 | .370 | .389 | .379 | .348 | .431 | .453 | .051 | .305 | .354 | .337 | .366 |
| | $M_{50}$ | .641 | .538 | .505 | .527 | .652 | .512 | .584 | .599 | .102 | .514 | .632 | .576 | .549 |
| | $M_{10}$ | .683 | .558 | .486 | .558 | .723 | .513 | .626 | .565 | .222 | .557 | .677 | .565 | .565 |
| **GEMINI-FLASH** *as Evaluator* | | | | | | | | | | | | | | |
| PAIRWISE | all | .531 | .505 | .500 | .509 | .519 | .473 | .538 | .574 | .415 | .471 | .519 | .531 | .509 |
| | filter | .574 | .540 | .517 | .520 | .557 | .444 | .561 | .629 | .410 | .373 | .556 | .565 | .529 |
| SCORE | $M_{100}$ | .482 | .433 | .433 | .362 | .521 | .368 | .497 | .455 | .453 | .377 | .560 | .429 | .420 |
| | $M_{50}$ | .642 | .584 | .510 | .517 | .661 | .497 | .630 | .616 | .528 | .510 | .674 | .600 | .567 |
| | $M_{10}$ | .746 | .606 | .504 | .551 | .719 | .483 | **.756** | .665 | **.800** | **.593** | **.817** | .576 | .599 |
| **GEMINI-PRO** *as Evaluator* | | | | | | | | | | | | | | |
| PAIRWISE | all | .539 | .538 | .493 | .513 | .558 | .435 | .409 | .554 | .396 | .437 | .522 | .530 | .509 |
| | filter | .598 | .588 | .547 | .564 | .628 | .471 | .557 | .658 | .423 | .412 | .560 | .569 | .557 |
| SCORE | $M_{100}$ | .494 | .442 | .450 | .397 | .555 | .361 | .483 | .476 | .434 | .363 | .460 | .417 | .428 |
| | $M_{50}$ | .653 | .612 | .551 | .520 | .692 | .520 | .600 | .631 | .566 | .511 | .567 | .560 | .581 |
| | $M_{10}$ | .742 | .695 | **.573** | .533 | .762 | .524 | .681 | .662 | .600 | .590 | .732 | .572 | .633 |
| **Aggregated Evaluator** | | | | | | | | | | | | | | |
| PAIRWISE | all | .525 | .515 | .482 | .509 | .526 | .477 | .508 | .542 | .445 | .484 | .495 | .520 | .502 |
| | filter | .577 | .574 | .498 | .489 | .579 | .487 | .549 | .600 | .457 | .416 | .531 | .528 | .524 |
| LOGITS | $M_{100}$ | .527 | .540 | .525 | .508 | .537 | .512 | .529 | .478 | .500 | .509 | .459 | .543 | .514 |
| | $M_{50}$ | .541 | .568 | .524 | .505 | .559 | .521 | .526 | .470 | .529 | .515 | .453 | .575 | .524 |
| | $M_{10}$ | .559 | .669 | .432 | .528 | .556 | .594 | .533 | .429 | .300 | .514 | .409 | .611 | .511 |
| SCORE | $M_{100}$ | .416 | .392 | .396 | .378 | .463 | .355 | .472 | .441 | .323 | .341 | .439 | .355 | .398 |
| | $M_{50}$ | .619 | .588 | .513 | .546 | .670 | .524 | .602 | .615 | .435 | .512 | .597 | .558 | .565 |
| | $M_{10}$ | .701 | .640 | .512 | .571 | .721 | .535 | .692 | .613 | .504 | .584 | .678 | .561 | .609 |

Table 9: Accuracy of wrong-over-wrong preference elicited from MISTRAL-7B, GEMINI-FLASH and GEMINI-PRO, and also the accuracy of heuristic, consistency-based, logits-based and score-based methods with a margin of $M_{100}$ that are not displayed in the main paper. $f^{(s)}_{GPT-4o}$ with $M_{10}$ still achieves the best weighted average $\text{Acc}_{WoW} = 0.709$ across all datasets.

| Method | Margin | LLAMA3-8B NLG | CP | GPT-3.5 NLG | CP | GPT-4O NLG | CP | Overall |
|---|---|---|---|---|---|---|---|---|
| | | *Model-independent* | | | | | | |
| HEURISTIC | $M_{100}$ | 0.510 | 0.506 | 0.472 | 0.504 | 0.535 | 0.481 | 0.501 |
| | $M_{50}$ | 0.509 | 0.506 | 0.468 | 0.519 | 0.547 | 0.464 | 0.502 |
| | $M_{10}$ | 0.457 | 0.512 | 0.476 | 0.501 | 0.486 | 0.475 | 0.478 |
| CONSISTENCY | $M_{100}$ | 0.359 | 0.281 | 0.323 | 0.300 | 0.269 | 0.301 | 0.318 |
| | $M_{50}$ | 0.544 | 0.478 | 0.514 | 0.499 | 0.432 | 0.498 | 0.505 |
| | $M_{10}$ | 0.509 | 0.535 | 0.506 | 0.492 | 0.302 | 0.468 | 0.478 |
| | | *LLaMA3-8B as Evaluator* | | | | | | |
| PAIRWISE | all | 0.501 | 0.516 | 0.477 | 0.493 | 0.551 | 0.504 | 0.503 |
| | filter | 0.586 | 0.560 | 0.538 | 0.504 | 0.500 | 0.510 | 0.545 |
| LOGITS | $M_{100}$ | 0.527 | 0.495 | 0.469 | 0.507 | 0.574 | 0.525 | 0.511 |
| | $M_{50}$ | 0.539 | 0.488 | 0.456 | 0.508 | 0.625 | 0.532 | 0.518 |
| | $M_{10}$ | 0.562 | 0.512 | 0.453 | 0.485 | 0.656 | 0.525 | 0.530 |
| SCORE | $M_{100}$ | 0.426 | 0.411 | 0.421 | 0.406 | 0.412 | 0.372 | 0.417 |
| | $M_{50}$ | 0.547 | 0.524 | 0.536 | 0.534 | 0.593 | 0.529 | 0.546 |
| | $M_{10}$ | 0.529 | 0.564 | 0.552 | 0.545 | 0.636 | 0.511 | 0.557 |
| | | *GPT-3.5 as Evaluator* | | | | | | |
| PAIRWISE | all | 0.505 | 0.485 | 0.514 | 0.469 | 0.507 | 0.499 | 0.502 |
| | filter | **0.802** | 0.547 | **0.590** | **0.585** | **0.882** | 0.535 | **0.642** |
| LOGITS | $M_{100}$ | - | - | 0.479 | 0.518 | - | - | 0.489 |
| | $M_{50}$ | - | - | 0.472 | 0.521 | - | - | 0.484 |
| | $M_{10}$ | - | - | 0.492 | 0.515 | - | - | 0.498 |
| SCORE | $M_{100}$ | 0.368 | 0.311 | 0.401 | 0.355 | 0.333 | 0.219 | 0.359 |
| | $M_{50}$ | 0.579 | 0.510 | 0.540 | 0.538 | 0.593 | 0.438 | 0.553 |
| | $M_{10}$ | 0.578 | 0.496 | 0.527 | 0.501 | 0.614 | **0.635** | 0.553 |
| | | *GPT-4o as Evaluator* | | | | | | |
| PAIRWISE | all | 0.560 | 0.495 | 0.554 | 0.512 | 0.575 | 0.503 | 0.546 |
| | filter | 0.586 | 0.519 | 0.569 | 0.520 | 0.613 | 0.512 | 0.568 |
| LOGITS | $M_{100}$ | - | - | - | - | 0.546 | 0.525 | 0.564 |
| | $M_{50}$ | - | - | - | - | 0.577 | 0.532 | 0.594 |
| | $M_{10}$ | - | - | - | - | 0.630 | 0.525 | 0.617 |
| SCORE | $M_{100}$ | 0.352 | 0.346 | 0.344 | 0.402 | 0.399 | 0.381 | 0.361 |
| | $M_{50}$ | 0.573 | 0.523 | 0.538 | 0.513 | 0.554 | 0.487 | 0.546 |
| | $M_{10}$ | 0.551 | **0.611** | 0.535 | 0.490 | 0.529 | 0.397 | 0.541 |

Table 10: Accuracy of wrong-over-wrong preference on NLGraph (full) and ChessPuzzle dataset. $f_{GPT\text{-}3.5}^{(p)}$ with consistency check achieve the best weighted average $\text{Acc}_{WoW} = 0.642$ across all datasets.

---

**Algorithm 1** $\mathcal{D}_{WoW}$ generation pipeline

---

**Input:** Original dataset $\mathcal{D}_0 = \{(\boldsymbol{q}^{(i)}, \boldsymbol{a}_{gt}^{(i)})\}_{i=1}^n$
$\mathcal{D}_{WoW} = \emptyset$
**for** $i$ in $1, 2, \ldots, n$ **do**
  Sample $m$ answers from LLM: $\{\boldsymbol{a}_j^{(i)}\}_{j=1}^m \sim \text{LLM}(\boldsymbol{q}^{(i)})$
  Filter out correct answers: $\{\boldsymbol{a}_j^{(i)} \mid \boldsymbol{a}_j^{(i)} \neq \boldsymbol{a}_{gt}^{(i)}\}_{j=1}^{m'}$
  **for** $\boldsymbol{a}_j^{(i)}, \boldsymbol{a}_k^{(i)}$ in combination($\{\boldsymbol{a}_j^{(i)}\}_{j=1}^{m'}$) **do**
    **if** $f_{LLM}(\boldsymbol{a}_j^{(i)} \succ \boldsymbol{a}_k^{(i)} \mid \boldsymbol{q}^{(i)}) = 1$ **then**
      $\boldsymbol{a}_c, \boldsymbol{a}_r = \boldsymbol{a}_j^{(i)}, \boldsymbol{a}_k^{(i)}$
    **else if** $f_{LLM}(\boldsymbol{a}_j^{(i)} \succ \boldsymbol{a}_k^{(i)} \mid \boldsymbol{q}^{(i)}) = -1$ **then**
      $\boldsymbol{a}_c, \boldsymbol{a}_r = \boldsymbol{a}_k^{(i)}, \boldsymbol{a}_j^{(i)}$
    **end if**
    Append $(\boldsymbol{q}^{(i)}, \boldsymbol{a}_c, \boldsymbol{a}_r)$ to $\mathcal{D}_{WoW}$
  **end for**
**end for**
**return** $\mathcal{D}_{WoW}$

| Margin | $3 \succ 0$ | $3 \succ 1$ | $2 \succ 0$ | $3 \succ 2$ | $2 \succ 1$ | $1 \succ 0$ |
|---|---|---|---|---|---|---|
| *logits (self)* | | | | | | |
| $M_{100}$ | **0.571** | 0.543 | 0.557 | 0.500 | 0.520 | 0.502 |
| $M_{50}$ | **0.590** | 0.553 | 0.551 | 0.514 | 0.533 | 0.531 |
| $M_{10}$ | **0.649** | 0.576 | 0.625 | 0.540 | 0.464 | 0.629 |
| *score (LLaMA3-8B)* | | | | | | |
| $M_{100}$ | **0.574** | 0.527 | 0.512 | 0.423 | 0.467 | 0.459 |
| $M_{50}$ | **0.749** | 0.711 | 0.658 | 0.541 | 0.610 | 0.587 |
| $M_{10}$ | 0.758 | **0.774** | 0.714 | 0.607 | 0.667 | 0.645 |
| *score (GPT-3.5)* | | | | | | |
| $M_{100}$ | **0.308** | 0.238 | 0.233 | 0.189 | 0.188 | 0.236 |
| $M_{50}$ | **0.463** | 0.454 | **0.463** | 0.399 | 0.392 | 0.431 |
| $M_{10}$ | **0.780** | 0.776 | 0.670 | 0.617 | 0.585 | 0.434 |
| *score (GPT-4o)* | | | | | | |
| $M_{100}$ | **0.753** | 0.668 | 0.575 | 0.509 | 0.481 | 0.339 |
| $M_{50}$ | **0.917** | 0.895 | 0.812 | 0.724 | 0.764 | 0.508 |
| $M_{10}$ | **0.958** | 0.940 | 0.829 | 0.792 | 0.867 | 0.529 |
| average | **0.696** | 0.665 | 0.607 | 0.533 | 0.558 | 0.463 |

Table 11: Preference accuracy for wrong-over-wrong pair with different wrongness margins. $a \succ b$ means preferring an answer filling in $a$ blanks correctly than an answer filling in $b$ blanks correctly. To have an overview, we also include correct answers which fill in 3 blanks correctly. We experiment on score-based eliciting with 3 LLM evaluators respectively and logits-based with self-evaluation. The results are an average on 3 generators. The largest wrongness margin $3 \succ 1$ yields the most accurate preference.

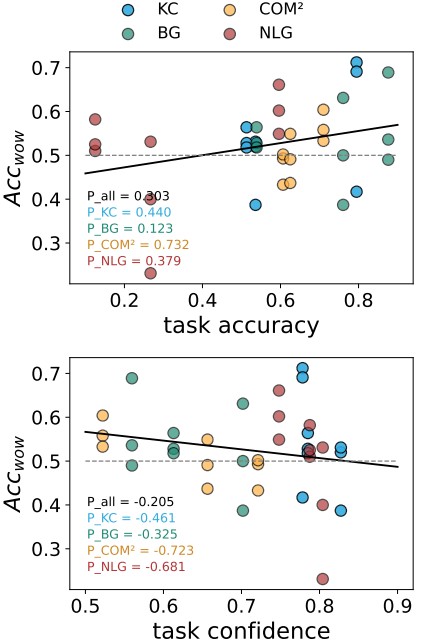

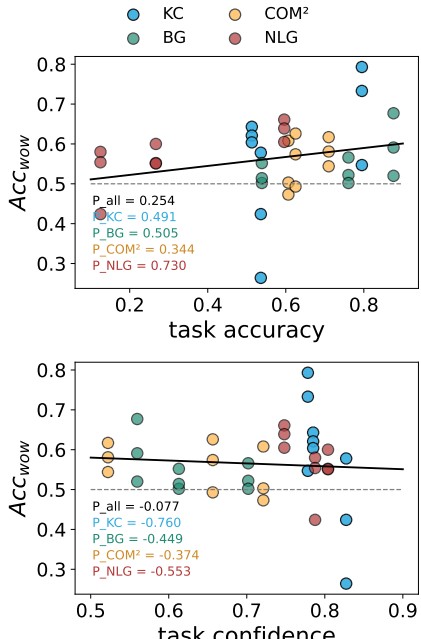

Figure 5: Correlation between task accuracy, confidence and $\text{Acc}_{WoW}$ for pairwise comparison with consistency check. Data points are from all 3 LLMs we used to elicit wong-over-wrong preferences. $P$ stands for Pearson correlation coefficient.

Figure 6: Correlation between task accuracy, confidence and $\text{Acc}_{WoW}$ for score-based with $M_{50}$. Data points are from all 3 LLMs we used to elicit wrong-over-wrong preferences. $P$ stands for Pearson correlation coefficient.

| Method | KC | | | | COM$^2$ | | | |
|---|---|---|---|---|---|---|---|---|
| | $P_{micro}$ | $P_{macro}$ | $R_{micro}$ | $F1_{micro}$ | $P_{micro}$ | $P_{macro}$ | $R_{micro}$ | $F1_{micro}$ |
| ORIGINAL | 0.564 | 0.561 | 0.559 | 0.559 | 0.689 | 0.679 | 0.682 | 0.676 |
| **SELF-GENERATOR** | | | | | | | | |
| PAIRWISE FILTER | **0.627** | **0.622** | **0.622** | **0.621** | 0.699 | **0.692** | **0.697** | **0.689** |
| SCORE $M_{50}$ | 0.597 | 0.598 | 0.591 | 0.592 | 0.675 | 0.663 | 0.666 | 0.659 |
| SCORE $M_{10}$ | 0.584 | 0.581 | 0.579 | 0.579 | 0.683 | 0.674 | 0.677 | 0.666 |
| ORACLE | 0.580 | 0.576 | 0.578 | 0.576 | **0.700** | 0.688 | 0.689 | 0.682 |
| **MIX-GENERATOR** | | | | | | | | |
| PAIRWISE FILTER | 0.574 | 0.572 | 0.569 | 0.570 | 0.698 | 0.687 | 0.692 | 0.686 |
| SCORE $M_{50}$ | 0.590 | 0.592 | 0.587 | 0.588 | 0.674 | 0.658 | 0.665 | 0.659 |
| SCORE $M_{10}$ | 0.565 | 0.563 | 0.562 | 0.562 | 0.680 | 0.671 | 0.676 | 0.666 |
| ORACLE | 0.581 | 0.579 | 0.580 | 0.577 | 0.699 | 0.684 | 0.691 | 0.684 |

Table 12: Precision, recall and F1 on Knowledge Crosswords and COM$^2$. Wrong-over-wrong alignment indeed can make LLMs generate more correct answers regardless of label distribution.

---

**Prompt:** You are a helpful assistant whose goal is to select the preferred AI model's output for a given instruction based on their correctness.
**Instruction:**
`[question here]`
**Output 1:**
`[answer ` $a_1$ ` here]`
**Output 2:**
`[answer ` $a_2$ ` here]`
Please first briefly describe your reasoning (in less than 100 words), and then conclude with "Preferred output: 1 or 2." in the last line. Let's think step-by-step.

---

Table 13: Prompt for pairwise comparison wrong-over-wrong preference eliciting.

---

**Prompt:** Below is an instruction from a user and five candidate responses. Please grade the correctness of each response on a 5-point scale from 0 to 5, where 0 is the lowest correctness, and 5 is the highest correctness. Only use integers for the ratings.
**Instruction:**
`[question here]`
**Response 1:**
`[answer ` $a_1$ ` here]`
**Response 2:**
`[answer ` $a_2$ ` here]`
**Response 3:**
`[answer ` $a_3$ ` here]`
**Response 4:**
`[answer ` $a_4$ ` here]`
**Response 5:**
`[answer ` $a_5$ ` here]`
Please first briefly describe your reasoning (in less than 100 words for each response), followed by a new line in the format of "Score: <rating>". Let's think step-by-step.

---

Table 14: Prompt for score-based wrong-over-wrong preference eliciting.

**Question:**
Instruction: Pick the correct answer for each blank that satisfies all the given constraints. Constraints: (Miles_Goodman, wroteMusicFor, blank 3); (Julien_Temple, directed, blank 1); (blank 2, actedIn, blank 3); (blank 2, actedIn, blank 1); (Jason_Alexander, actedIn, blank 3);

**Correct answer:**
blank 1: Earth_Girls_Are_Easy, blank 2: Damon_Wayans, blank 3: Blankman

**Original answer:**
Let's analyze the question step by step: 1. Julien Temple directed blank 1. From the options, we can conclude that blank 1 is either "Earth_Girls_Are_Easy" or "Burn_'Em_Up_Barnes". 2. Miles Goodman wrote music for blank 3. From the options, we can conclude that blank 3 is either "Blankman" or "On_Guard_(1997_film)". 3. blank 2 acted in blank 1 and blank 3. From the options, we can conclude that blank 2 is the same person in both cases. The only options that satisfy this constraint are B and D. 4. Jason Alexander acted in blank 3. From the options, we can conclude that blank 3 is either "Blankman" or "On_Guard_(1997_film)". Given the previous conclusion, blank 3 must be "Blankman". 5. Putting it all together, we can conclude that the correct answer is B. blank 1: Burn_'Em_Up_Barnes, blank 2: Justin_Theroux, blank 3: Blankman.
Final answer: B.

**Less wrong answer:**
Let's analyze the question step by step: 1. Julien_Temple, directed, blank 1: Julien Temple directed Earth_Girls_Are_Easy, so blank 1 must be Earth_Girls_Are_Easy. 2. blank 2, actedIn, blank 1: From step 1, blank 1 is Earth_Girls_Are_Easy. So, blank 2 must have acted in Earth_Girls_Are_Easy. Let's call blank 2 X. 3. blank 2, actedIn, blank 3: From step 2, X acted in Earth_Girls_Are_Easy and also acted in blank 3. 4. Jason_Alexander, actedIn, blank 3: Jason Alexander acted in blank 3. 5. blank 2, actedIn, blank 1: From step 2, X acted in Earth_Girls_Are_Easy again (because it's the same film). 6. Miles_Goodman, wroteMusicFor, blank 3: Miles Goodman wrote music for blank 3. 7. Julien_Temple, directed, blank 1: We already knew this, but it's repeated to ensure consistency.
Now, let's analyze the options: A. blank 1: Burn_'Em_Up_Barnes, blank 2: Justin_Theroux, blank 3: On_Guard_(1997_film): Burn_'Em_Up_Barnes is not directed by Julien Temple, and Justin Theroux didn't act in On_Guard_(1997_film). Discarded. B. blank 1: Earth_Girls_Are_Easy, blank 2: Damon_Wayans, blank 3: Blankman: We already know that blank 1 is Earth_Girls_Are_Easy. Damon Wayans didn't act in Earth_Girls_Are_Easy, and Blankman isn't the same film as On_Guard_(1997_film). Discarded. C. blank 1: Burn_'Em_Up_Barnes, blank 2: Justin_Theroux, blank 3: Blankman: Burn_'Em_Up_Barnes is not directed by Julien Temple, and Justin Theroux didn't act in Blankman. Discarded. D. blank 1: Earth_Girls_Are_Easy, blank 2: Justin_Theroux, blank 3: Blankman: This option fits all the constraints: - blank 1 is Earth_Girls_Are_Easy (step 1) - blank 2 is Justin_Theroux (step 2) - blank 3 is Blankman (works with steps 3, 4, and 6)
So, the final answer is: D. blank 1: Earth_Girls_Are_Easy, blank 2: Justin_Theroux, blank 3: Blankman
(Correctly filled blanks 1 → 2)

**More correct answer:**
Let's break down the constraints step by step: 1. (Miles_Goodman, wroteMusicFor, blank 3) This constraint tells us that Miles Goodman wrote music for a movie, and the title of that movie is blank 3. 2. (Julien_Temple, directed, blank 1) This constraint tells us that Julien Temple directed a movie, and the title of that movie is blank 1. 3. (blank 2, actedIn, blank 3) This constraint tells us that blank 2 acted in a movie, and the title of that movie is blank 3. 4. (blank 2, actedIn, blank 1) This constraint tells us that blank 2 acted in another movie, and the title of that movie is blank 1. 5. (Jason_Alexander, actedIn, blank 3) This constraint tells us that Jason Alexander acted in a movie, and the title of that movie is blank 3.

Table 15: Examples showing the effect of wrong-over-wrong alignment on Knowledge Crosswords dataset (part 1).

Now, let's analyze the options: A. blank 1: Burn_'Em_Up_Barnes, blank 2: Justin_Theroux, blank 3: Blankman Burn_'Em_Up_Barnes is not a well-known movie, and Blankman is not a movie that was directed by Julien Temple or written music for by Miles Goodman. B. blank 1: Earth_Girls_Are_Easy, blank 2: Justin_Theroux, blank 3: On_Guard_(1997_film) Earth_Girls_Are_Easy is a real movie, but On_Guard_(1997_film) is not a movie that was directed by Julien Temple or written music for by Miles Goodman. Additionally, Justin Theroux did not act in On_Guard_(1997_film). C. blank 1: Earth_Girls_Are_Easy, blank 2: Damon_Wayans, blank 3: Blankman Damon Wayans is an actor, but he did not act in Earth_Girls_Are_Easy.

The only option that satisfies all the constraints is: C. blank 1: Earth_Girls_Are_Easy, blank 2: Damon_Wayans, blank 3: Earth_Girls_Are_Easy This option fills in the blanks as follows: - Julien Temple directed Earth_Girls_Are_Easy (constraint 2) - Damon Wayans acted in Earth_Girls_Are_Easy (constraint 4) - Damon Wayans acted in Earth_Girls_Are_Easy (constraint 3) - Jason Alexander acted in Earth_Girls_Are_Easy (constraint 5) - Miles Goodman wrote music for Earth_Girls_Are_Easy (constraint 1)

The movie Earth_Girls_Are_Easy is a 1988 musical comedy film directed by Martin Davidson, and Damon Wayans and Jason Alexander both appeared in the film. Miles Goodman wrote the original songs for the movie.

(Correctly filled blanks 1 → 3)

**Better calibrated answer:**
Let's analyze the question step by step: 1. (Julien_Temple, directed, blank 1) - This constraint states that Julien Temple directed a movie, and the title of that movie is blank 1. 2. (blank 2, actedIn, blank 1) - This constraint states that blank 2 acted in the same movie as in step 1. 3. (blank 2, actedIn, blank 3) - This constraint states that blank 2 acted in another movie, and the title of that movie is blank 3. 4. (Miles_Goodman, wroteMusicFor, blank 3) - This constraint states that Miles Goodman wrote music for the same movie as in step 3. 5. (Jason_Alexander, actedIn, blank 3) - This constraint states that Jason Alexander acted in the same movie as in step 3.

Now, let's analyze the options: A. blank 1: Earth_Girls_Are_Easy, blank 2: Damon_Wayans, blank 3: Blankman - This option is incorrect because Earth_Girls_Are_Easy is a 1984 film directed by Sidney Lumet, not Julien Temple. Additionally, Justin Theroux was not involved in that film. B. blank 1: Burn_'Em_Up_Barnes, blank 2: Justin_Theroux, blank 3: Blankman - This option is incorrect because Justin Theroux was not involved in the 1984 film Burn_'Em_Up_Barnes, which is a TV movie. Additionally, Jason Alexander was not involved in that film. C. blank 1: Burn_'Em_Up_Barnes, blank 2: Justin_Theroux, blank 3: On_Guard_(1997_film) - This option is incorrect because Jason Alexander was not involved in the 1997 film On_Guard_, which is a TV movie. Additionally, Miles Goodman was not involved in that film. D. blank 1: Earth_Girls_Are_Easy, blank 2: Justin_Theroux, blank 3: Blankman - This option is incorrect because Justin Theroux was not involved in the 1984 film Earth_Girls_Are_Easy, and Jason Alexander was not involved in that film.

After analyzing the options, the only remaining possibility is: B. blank 1: Burn_'Em_Up_Barnes, blank 2: Justin_Theroux, blank 3: Blankman This option satisfies all the constraints: - Julien Temple directed the movie Burn_'Em_Up_Barnes (1990), which is blank 1. - Justin Theroux acted in Burn_'Em_Up_Barnes, which is blank 2. - Justin Theroux also acted in Blankman (1994), which is blank 3, along with Jason Alexander. - Miles Goodman wrote music for Blankman, which is blank 3.

Therefore, the final answer is: B. blank 1: Burn_'Em_Up_Barnes, blank 2: Justin_Theroux, blank 3: Blankman

(Even the answers are equally wrong, NLL increases 0.129 → 0.185)

Table 16: Examples showing the effect of wrong-over-wrong alignment on Knowledge Croswords dataset (part 2).

**Failed answer:**

Let's break down the constraints step by step: 1. (Miles_Goodman, wroteMusicFor, blank 3): This constraint tells us that Miles Goodman wrote music for a movie, and the title of that movie is blank 3. From the options, we can see that only options C and D mention Miles Goodman writing music. 2. (Julien_Temple, directed, blank 1): This constraint tells us that Julien Temple directed a movie, and the title of that movie is blank 1. From the options, we can see that only options B and C mention Julien Temple directing a movie. The title of that movie is either Burn_'Em_Up_Barnes or Earth_Girls_Are_Easy. 3. (blank 2, actedIn, blank 3): This constraint tells us that blank 2 acted in a movie, and the title of that movie is blank 3. From the previous step, we know that blank 1 is either Burn_'Em_Up_Barnes or Earth_Girls_Are_Easy. Since blank 2 is the same in both options B and C, we can conclude that blank 2 acted in a movie with the title blank 3, which is either Burn_'Em_Up_Barnes or Earth_Girls_Are_Easy. 4. (blank 2, actedIn, blank 1): This constraint is redundant, as it's already known that blank 2 acted in a movie with the title blank 1, which is either Burn_'Em_Up_Barnes or Earth_Girls_Are_Easy. 5. (Jason_Alexander, actedIn, blank 3): This constraint tells us that Jason Alexander acted in a movie, and the title of that movie is blank 3. From the previous steps, we know that blank 3 is either Burn_'Em_Up_Barnes or Earth_Girls_Are_Easy. Since Jason Alexander acted in one of those movies, we can conclude that Jason Alexander acted in a movie with the title blank 3.

Now, let's analyze the remaining constraints: - Justin Theroux needs to be blank 2, as it's mentioned in constraints 3 and 4. - Blank 3 needs to be either Burn_'Em_Up_Barnes or Earth_Girls_Are_Easy, as it's mentioned in constraints 1, 3, and 5. - Blank 1 needs to be either Burn_'Em_Up_Barnes or Earth_Girls_Are_Easy, as it's mentioned in constraints 2 and 4.

The only option that satisfies all the constraints is: B. blank 1: Burn_'Em_Up_Barnes, blank 2: Justin_Theroux, blank 3: Earth_Girls_Are_Easy This option satisfies all the constraints: - Miles Goodman wrote music for Earth_Girls_Are_Easy. - Julien Temple directed Burn_'Em_Up_Barnes. - Justin Theroux acted in both Burn_'Em_Up_Barnes and Earth_Girls_Are_Easy. - Jason Alexander acted in Earth_Girls_Are_Easy.

(Correctly filled blanks 1 → 0)

Table 17: Examples showing the effect of wrong-over-wrong alignment on Knowledge Crosswords dataset (part 3).

---

**Question:**
Tell me a bio of Luiz Inácio Lula da Silva.

**Original answer:**
Luiz Inácio Lula da Silva, commonly known as Lula, is a Brazilian politician who served as the 35th President of Brazil from 2003 to 2011. He is one of the most popular and influential politicians in Brazil's recent history. Early Life: Lula was born on October 27, 1945, in Garanhuns, a small town in the northeastern state of Pernambuco, Brazil. His father was a coffee farmer and his mother was a seamstress. He was the fourth of seven children, and his family struggled financially throughout his childhood. Lula grew up in a poor and rural area, where he learned the value of hard work and the importance of education. He dropped out of school at the age of 12 to work in a textile factory, where he eventually became a union leader. This early experience in labor organizing would shape his future political career. Political Career: Lula's political career began in the late 1970s, when he became involved with the Brazilian Workers' Party (PT), a left-wing party founded in the 1980s. He quickly rose through the ranks of the party, becoming its presidential candidate in 1989. Although Lula lost the election to Fernando Collor de Mello, he continued to build his political base and gained popularity for his fight against corruption and social inequality. In 1994, he was accused of corruption and money laundering, which led to a conviction and a four-year prison sentence. However, in 1998, the Supreme Court overturned Lula's conviction, citing lack of evidence. This cleared the way for him to run for president again in 2002. He won the election in a landslide, receiving nearly 65Presidency: During his presidency, Lula implemented a series of reforms aimed at reducing poverty and inequality. He increased social spending, increased the minimum wage, and expanded access to education and healthcare. He also launched a number of economic initiatives, including a debt reduction plan and an infrastructure development program. Lula's presidency was marked by strong economic growth, which lifted millions of Brazilians out of poverty. He also gained international attention for his leadership on issues such as climate change and social justice. Presidential Terms: Lula served two consecutive terms as president, from 2003 to 2011. During his first term, he focused on addressing poverty and inequality, and implemented a range of social welfare programs. During his second term, Lula emphasized economic development and infrastructure projects. He invested heavily in public transportation, energy, and telecommunications, and launched a series of initiatives to promote sustainable development. Investigation and Charges: After leaving office in 2011, Lula was investigated for corruption and money laundering. He was accused of receiving bribes from a construction company, OAS, in exchange for favors and contracts. He denied the allegations, but was arrested and charged in 2018. In 2019, Lula was convicted and sentenced to nine years and six months in prison. He maintained his innocence and appealed the conviction, questioning the fairness of the trial. Return to Politics: Despite his imprisonment, Lula remained a beloved figure in Brazil, and his popularity continued to grow. In 2021, the Supreme Court ordered his release from prison, citing the unfairness of the trial. Lula was subsequently reinstated as a senator and began to reorganize the Workers' Party. In 2022, Lula announced his candidacy for the 2022 Brazilian presidential election, and won the election in a landslide. He is currently serving as the 39th President of Brazil, having taken office on January 1, 2023. Legacy: Luiz Inácio Lula da Silva's legacy is complex and multifaceted. On the one hand, he is credited with reducing poverty and inequality, and promoting social justice and economic development in Brazil. On the other hand, his presidency was marked by corruption scandals, and his own conviction and imprisonment have raised questions about his integrity and accountability. Despite these controversies, Lula remains a towering figure in Brazilian politics, and his commitment to social justice and poverty reduction has inspired generations of Brazilians. His political career is a testament to the power of grassroots organizing and the importance of fighting for social justice, even in the face of significant challenges and setbacks.

---

Table 18: Examples showing the effect of wrong-over-wrong alignment on Bio Generation dataset (part 1).

**Less wrong / More correct / Better calibrated answer:**

Luiz Inácio Lula da Silva, commonly known as Lula, is a Brazilian politician and former trade unionist who served as the 34th President of Brazil from 2003 to 2011. He was born on October 27, 1945, in Garanhuns, a small town in the state of Pernambuco, Brazil. Lula's early life was marked by poverty and hardship. His parents were both poor and separated when Lula was young, and he was forced to work at a young age to support his family. He eventually moved to São Paulo and became involved in trade union activism, eventually becoming one of the key figures in the Brazilian Workers' Central Union (CUT). Lula's political career began in the 1980s, when he was elected as the leader of the Metalworkers' Union and became a prominent figure in the Brazilian labor movement. He was one of the founders of the Workers' Party (PT) in 1980, and he quickly became the party's most prominent figure. In 2002, Lula became the first worker and the first non-PSDB (Brazilian Social Democracy Party) candidate to run for President of Brazil. Despite initial skepticism about his candidacy, he won the election in a landslide, receiving over 61During his two terms in office, Lula implemented a number of significant reforms and policies. He implemented a series of social and economic programs aimed at reducing poverty and income inequality, including the Bolsa Família program, which provided financial assistance to low-income families, and the Fome Zero program, which aimed to eradicate hunger and poverty. Lula also played a key role in shaping Brazil's foreign policy, particularly in the areas of trade and international relations. He was one of the key figures in the formation of the BRICS (Brazil, Russia, India, China, and South Africa) alliance, and he played a significant role in shaping the policy of the G20 and the United Nations. After leaving office in 2011, Lula remained a prominent figure in Brazilian politics and continued to play a significant role in shaping his party's policies and candidates. In 2018, he was elected as the Chair of the PT, and he became the leading figure in the country's political opposition, advocating for changes to the country's political system and advocating for the nomination of his chosen successor, Fernando Haddad, in the 2018 presidential election. In 2018, Lula's candidacy in the presidential election was barred due to a corruption conviction, and he was replaced as the PT's presidential candidate by Fernando Haddad. However, Haddad ultimately lost the election to Jair Bolsonaro. Lula's legacy in Brazil is complex and controversial. On one hand, he is widely credited with improving the lives of millions of Brazilians, particularly the poor and the working class, through his social and economic policies. On the other hand, he was also accused of corruption and was ultimately convicted of accepting bribes and ordered to serve a prison sentence. He is currently appealing his conviction and is being held in a federal prison in Brazil. Despite his current legal situation, Lula's political career and legacy continue to be subject to significant debate and discussion in Brazil and around the world. His presidency marked a significant shift in the country's political landscape, and his policies and legacy continue to have a significant impact on Brazilian politics and society.

(FActScore improves 0.836 → 0.925, NLL drops 0.512 → 0.416)

Table 19: Examples showing the effect of wrong-over-wrong alignment on Bio Generation dataset (part 2).

**Failed answer:**

Luiz Inácio Lula da Silva, commonly known as Lula, is a Brazilian politician and former trade unionist who served as the 34th President of Brazil from 2003 to 2011 and again from 2016 to 2018. Early Life and Career: Lula was born on October 11, 1945, in Garanhuns, Pernambuco, Brazil, to Armando da Silva and Hermilda da Silva. His early life was marked by poverty and hardship, with his family struggling to make ends meet. Lula's mother died of complications related to a gas explosion when he was just 10 years old, and he was subsequently raised by his aunt. Lula's early career was marked by his involvement in the trade union movement. He became involved with the Metal Workers Union, where he quickly rose through the ranks and became one of the most prominent union leaders in Brazil. During this time, he became known for his charismatic leadership style, his ability to connect with ordinary people, and his advocacy for workers' rights. Political Career: In 1980, Lula co-founded the Workers' Party (PT), a left-wing political party that sought to represent the interests of working-class Brazilians. He became the party's first president and held the position until 2002. In 2002, Lula became the first trade unionist to run for and win the presidency of Brazil, defeating the incumbent Fernando Henrique Cardoso. He was re-elected in 2006 and served two terms as president until 2011. During his first tenure as president, Lula implemented a range of policies aimed at reducing poverty and income inequality in Brazil. He increased access to education and healthcare, and implemented policies to stimulate economic growth and job creation. In 2010, Lula was succeeded by his former minister and protégé, Dilma Rousseff, who became the first female president of Brazil. However, Rousseff's presidency was later marred by corruption scandals and political instability, culminating in her impeachment and removal from office in 2016. Lula returned to politics and ran again for the presidency in 2014, but was not eligible to run due to a constitutional ban on former presidents from holding the office again. He subsequently threw his support behind Rousseff's re-election bid, but she was ultimately impeached and removed from office. In 2018, Lula ran for the presidency again and won the election, becoming the first and only person to be re-elected as President of Brazil. However, his presidency was short-lived, as he was impeached and removed from office in August 2016, citing corruption allegations and constitutional violations. Lula was subsequently replaced by his vice president, Michel Temer. Personal Life: Lula has been married to Maria de Deus Inácio da Silva (known as Marinha) since 1970, and they have four sons together. In 2018, Lula was sentenced to 12 years and 1 month in prison for corruption and money laundering, stemming from the "Operation Car Wash" corruption scandal. He has since remained in jail, and his legacy and political future remain uncertain. Legacy: Lula's presidency was marked by significant economic and social changes in Brazil, including a reduction in poverty and income inequality, as well as an unprecedented level of social mobility. He is widely regarded as one of the most important figures in Brazil's recent history, and his legacy continues to shape Brazilian politics and society. Despite his political setbacks and legal troubles, Lula remains a powerful and influential figure in Brazilian politics, and his name continues to be mentioned as a potential candidate for future elections.

(FActScore drops 0.836 → 0.770)

Table 20: Examples showing the effect of wrong-over-wrong alignment on Bio Generation dataset (part 3).

---

**Question:**
Instruction: As an expert in commonsense reasoning, your task is to provide a concise response to a question based on the given context. The question focuses on studying the causes, effects, or attributes of personas related to the given context. Context: Noah had been planning a surprise visit for Everly. As part of the surprise, Noah buys a gift for Everly. When Everly arrives at the airport, Noah picks Everly up. Question: What event or state is both what Everly wants to do after Noah buys a gift for Everly and also what Everly wants to do after Noah picks Everly up at the airport?

**Correct answer:**
Ranking by commonsense plausibility: Everly says thank you $\succ$ Everly is shocked $\succ$ Everly go out to eat $\succ$ Everly rest
(The commonsense plausibility is given by Vera.)

**Original answer:**
Here's the step-by-step analysis: 1. The question asks about an event or state that is common to both scenarios: after Noah buys a gift for Everly and after Noah picks Everly up at the airport. 2. The context suggests that Noah is planning a surprise visit for Everly, and buying a gift is part of the surprise. 3. When Noah buys a gift, it implies that Everly didn't expect the gift, and therefore, will likely be surprised when she receives it. 4. When Noah picks Everly up at the airport, it's likely that Everly didn't expect the pick-up, and therefore, will also be surprised. 5. The common event or state that arises from both scenarios is Everly's surprise. Therefore, the correct answer is B. Everly is shocked.

**Less wrong answer:**
Let's break down the question step by step: 1. The question asks about an event or state that Everly wants to do after Noah buys a gift for her and also after Noah picks her up at the airport. 2. We know that Everly has been anticipating a surprise visit from Noah, which suggests that she's looking forward to seeing Noah. 3. When Noah buys a gift for Everly, it's likely that she wants to see or meet Noah to receive the gift, which means she wants to be with Noah. 4. Similarly, when Noah picks Everly up at the airport, she likely wants to be with Noah, as she was expecting a visit from him. 5. Therefore, the event or state that Everly wants to do after Noah buys a gift for her and after Noah picks her up at the airport is likely to be: **Being with Noah** Since options B-D do not seem related to this event, the correct answer is not present in the options. If I had to choose, I would say that the correct answer is not present in the options (**N/A**).
(Instead of giving a wrong answer, LLM choose to abstain.)

**More correct answer:**
Let's break down the question step by step: 1. The question asks about an event or state that is common to both situations: after Noah buys a gift for Everly and after Noah picks Everly up at the airport. 2. In both situations, Everly is receiving a service or action from Noah (a gift or transportation). 3. Given the context, it's likely that Everly wants to reciprocate or respond positively to Noah's actions. 4. Among the options, only one relates to Everly's response to Noah's actions: A. Everly says thank you. 5. Therefore, the most likely answer is A. Everly says thank you, as it is a common outcome for Everly after receiving a gift or being picked up by Noah. Final answer: A. Everly says thank you.
(LLM gives a right answer.)

---

Table 21: Examples showing the effect of wrong-over-wrong alignment on COM$^2$ dataset (part 1).

**Better calibrated answer:**
Let's break down the question step by step: 1. The question asks about an event or state that is common to two scenarios: after Noah buys a gift for Everly and after Noah picks Everly up at the airport. 2. In both scenarios, Everly is expected to be pleased or surprised, given the context of the surprise visit and gift. 3. The correct answer must capture this idea of Everly's positive reaction. 4. Options A, B, and D are not directly related to Everly's actions or reactions. 5. Only option C, "Everly rests", is too vague to be relevant to the context. Final answer: None of the options fit the description. The correct answer is not among the provided options.
(Even the answers are equally wrong, NLL increases $0.412 \rightarrow 0.509$)

**Failed answer:** Let's break down the question step by step: 1. The question asks about an event or state that Everly wants to do after two specific events: Noah buying a gift and Noah picking Everly up at the airport. 2. The question stem "What event or state..." suggests that we're looking for a common action or experience that Everly desires in both scenarios. 3. From the context, we know that Noah is planning a surprise visit and has bought a gift for Everly. This suggests that Everly is likely expecting a pleasant experience or a reunion with Noah. 4. After Noah picks Everly up at the airport, it's natural to infer that Everly wants to reunite with or see Noah, which is a pleasing experience. 5. Similarly, after Noah buys a gift for Everly, it's likely that Everly wants to see or have the gift, which is also a pleasing experience. Given this analysis, the correct answer is: C. Everly visits/see Noah. In both scenarios, Everly wants to experience the pleasure of being with or seeing Noah, which is the most likely common action or state that Everly desires.
(LLM gives an answer that doesn't appear in options)

Table 22: Examples showing the effect of wrong-over-wrong alignment on $COM^2$ dataset (part 2).

**Question:**
In an undirected graph, the nodes are numbered from 0 to 5, and the edges are: an edge between node 0 and node 2 with weight 2, an edge between node 0 and node 5 with weight 4, an edge between node 0 and node 3 with weight 1, an edge between node 0 and node 1 with weight 3, an edge between node 1 and node 2 with weight 4, an edge between node 1 and node 5 with weight 1, an edge between node 1 and node 4 with weight 4, an edge between node 2 and node 4 with weight 3, an edge between node 2 and node 3 with weight 1, an edge between node 3 and node 5 with weight 2, an edge between node 3 and node 4 with weight 3, an edge between node 4 and node 5 with weight 2. Q: Give the shortest path from node 0 to node 4. Please also give the total weight of the shortest path.

**Correct answer:**
$4 \; [0 \to 3 \to 4]$

**Original answer:**
Here's the step-by-step analysis: 1. Create a graph with nodes from 0 to 5 and the given edges with their corresponding weights. 2. Choose node 0 as the starting node and node 4 as the target node. 3. Run Dijkstra's algorithm to find the shortest path from node 0 to node 4. Here are the steps of Dijkstra's algorithm:

- Initialize a distance array with all distances set to infinity, except for node 0 which is set to 0. The distance array represents the shortest distance from node 0 to each node.

- Create a priority queue and add node 0 with a distance of 0. The priority queue is used to select the node with the shortest distance.

- While the priority queue is not empty, repeat the following steps: 1. Extract the node with the shortest distance from the priority queue. 2. For each neighbor of the extracted node, calculate the tentative distance as the distance to the extracted node plus the edge weight between them. 3. If the tentative distance is less than the current distance to the neighbor, update the distance array and add the neighbor to the priority queue.

The shortest path from node 0 to node 4 is the path with the minimum distance from node 0 to node 4. The shortest path from node 0 to node 4 is: $0 \to 2 \to 3 \to 4$ The total weight of the shortest path is: 2 + 1 + 3 = 6. Therefore, the final answer is: Shortest path: $0 \to 2 \to 3 \to 4$, Total weight: 6.

**Less wrong answer:**
Here's the step-by-step analysis:

1. Start at node 0.

2. Consider the edges from node 0: (0, 2) with weight 2, (0, 3) with weight 1, (0, 1) with weight 3, (0, 5) with weight 4. We'll choose the edge with the smallest weight, which is (0, 3) with weight 1.

3. Move to node 3.

4. Consider the edges from node 3: (3, 4) with weight 3, (3, 5) with weight 2, (3, 2) with weight 1. We'll choose the edge with the smallest weight, which is (3, 2) with weight 1.

5. Move to node 2.

6. Consider the edges from node 2: (2, 4) with weight 3. We'll choose this edge, since it's the only option.

7. Move to node 4.

The shortest path is: $0 \to 3 \to 2 \to 4$. The total weight of the shortest path is: $1 + 1 + 3 = 5$. Therefore, the answer is: The shortest path from node 0 to node 4 is $0 \to 3 \to 2 \to 4$, with a total weight of 5.
(The path found is shorter: $6 \to 5$)

Table 23: Examples showing the effect of wrong-over-wrong alignment on NLGraph dataset (part 1).

**More correct answer:** Let's analyze the question step by step:

1. We are given an undirected graph with nodes numbered from 0 to 5, and edges with weights between these nodes.

2. We want to find the shortest path from node 0 to node 4.

3. We can use a simple algorithm to find the shortest path:
   - Create a list of nodes to visit (initially, only node 0 is in the list).
   - For each node in the list, check if it's a neighbor of node 0 (i.e., if an edge exists).
   - If it's a neighbor, check the weight of the edge and add the neighbor to the list.
   - Mark the neighbor as visited.
   - Repeat until node 4 is reached.
   - The path from node 0 to node 4 is the sequence of nodes in the list, and the total weight is the sum of the edge weights along this path.

Applying this algorithm:Node $0 \rightarrow$ Node 3 (edge weight 1) $\rightarrow$ Node 4 (edge weight 3). The shortest path is: Node $0 \rightarrow$ Node $3 \rightarrow$ Node 4. The total weight is: 1 (from Node 0 to Node 3) + 3 (from Node 3 to Node 4) = 4. Final answer: The shortest path from node 0 to node 4 is Node $0 \rightarrow$ Node $3 \rightarrow$ Node 4, with a total weight of 4.
(Find the shortest path)

**Better calibrated answer:**
Here's the step-by-step analysis:

1. Create a graph from the given information. The graph looks like this:

2. Initialize a distance array with all distances set to infinity, except for the source node (node 0) which is set to 0:
$$\text{dist}[0] = 0$$
$$\text{dist}[1] = \infty$$
$$\text{dist}[2] = \infty$$
$$\text{dist}[3] = \infty$$
$$\text{dist}[4] = \infty$$
$$\text{dist}[5] = \infty$$

3. Relax all edges in the graph. This involves updating the distance array by considering all edges:

$$\text{dist}[1] = 3 \quad (0 \rightarrow 1 \text{ with weight } 3)$$
$$\text{dist}[2] = 2 \quad (0 \rightarrow 2 \text{ with weight } 2)$$
$$\text{dist}[3] = 1 \quad (0 \rightarrow 3 \text{ with weight } 1)$$
$$\text{dist}[4] = \infty$$
$$\text{dist}[5] = 4 \quad (0 \rightarrow 5 \text{ with weight } 4)$$
$$\text{dist}[2] = 3 \quad (1 \rightarrow 2 \text{ with weight } 4, \text{ but previously updated to } 2, \text{ so no change})$$
$$\text{dist}[3] = 3 \quad (1 \rightarrow 3 \text{ with weight } 4, \text{ but previously updated to } 1, \text{ so no change})$$
$$\text{dist}[3] = 2 \quad (2 \rightarrow 3 \text{ with weight } 1)$$
$$\text{dist}[4] = \infty$$
$$\text{dist}[5] = 5 \quad (1 \rightarrow 5 \text{ with weight } 1, \text{ then } 4, \text{ so no change})$$
$$\text{dist}[4] = 5 \quad (2 \rightarrow 4 \text{ with weight } 3)$$

4. The minimum distance from node 0 to node 4 is 5. The shortest path is $0 \rightarrow 2 \rightarrow 3 \rightarrow 4$.

Table 24: Examples showing the effect of wrong-over-wrong alignment on NLGraph dataset (part 2).

The total weight of the shortest path is:$0 \rightarrow 2$ (weight 2) $\rightarrow 2 \rightarrow 3$ (weight 1) $\rightarrow 3 \rightarrow 4$ (weight 3). Total weight: $2 + 1 + 3 = 6$.

(Even the answers are equally wrong, NLL increases $0.255 \rightarrow 0.373$)

**Failed answer:**

Let's break down the question step by step:

1. Identify the nodes connected to node 0: 1, 2, 3, 5

2. Weigh the edges connected to node 0:
   - Edge to node 1: weight 3
   - Edge to node 2: weight 2
   - Edge to node 3: weight 1
   - Edge to node 5: weight 4

3. Identify the nodes connected to node 1: 2, 4, 5
   - Edge to node 2: weight 4
   - Edge to node 4: weight 4
   - Edge to node 5: weight 1

4. The shortest path from node 0 to node 4 is through node 3:Node 0 $\rightarrow$ Node 3 $\rightarrow$ Node 4 (weights $1 \rightarrow 3$)

5. The total weight of the shortest path is:$0 \rightarrow 3$ (weight 1) $\rightarrow 3 \rightarrow 4$ (weight 3) $= 4$

The shortest path from node 0 to node 4 is Node 0 $\rightarrow$ Node 3 $\rightarrow$ Node 4, with a total weight of 4.

(Fail to calculate total weight correctly)

Table 25: Examples showing the effect of wrong-over-wrong alignment on NLGraph dataset (part 3).

