# OpenReview forum: "Varying Shades of Wrong: Aligning LLMs with Wrong Answers Only"
_ICLR.cc/2025/Conference — ICLR 2025 Poster_

### Official Review · Reviewer_AQ5T · 2024-10-31

**Soundness:** 3
**Presentation:** 3
**Contribution:** 3
**Rating:** 6
**Confidence:** 4

**Summary:**

This article introduces an approach based on wrong-over-wrong preference for aligning Large Language Models to address the issue of high costs associated with obtaining ground truth in this field.

**Strengths:**

This idea is interesting and beneficial for the entire field.

The authors conducted experiments using several popular LLMs across multiple datasets. The results indicate that utilizing wrong-over-wrong preferences can lead to LLMs generating fewer incorrect answers.

**Weaknesses:**

In the experiments, the authors initially employed a proxy function to compute correctness scores based on ground truth, and then derived wrong-over-wrong preferences from these scores. I believe this process contradicts the motivation behind this work, as the resulting wrong-over-wrong preferences do not significantly differ from traditional right-over-wrong preferences obtained through similar methods.

The authors primarily relied on accuracy metrics. However, due to the lack of calibration mitigation in the baselines, these metrics are influenced by label distribution.

The baselines were limited to the original models, without comparison against some state-of-the-art (SOTA) alignment methods.

**Questions:**

see Weaknesses

---

> ### Author Response · Authors · 2024-11-15
> **Author Response**
>
> We would like to thank the reviewer for their thoughtful comments and feedback.
>
> > In the experiments, the authors initially employed a proxy function to compute correctness scores based on ground truth, and then derived wrong-over-wrong preferences from these scores. I believe this process contradicts the motivation behind this work, as the resulting wrong-over-wrong preferences do not significantly differ from traditional right-over-wrong preferences obtained through similar methods.
>
> The core thing is: **wrong-over-wrong preferences are elicited from LLMs and not with proxy scores**. Proxy scores are employed to evaluate LLM-generated preferences: they only serve evaluation purposes as stated in the footnote of Page 2.
>
> The process essentially is: take a question, LLM sample multiple answers, LLM self-judge wrongness preferences among answers, align with these preferences, and then evaluate with the proxy function. In other words, aligning with only wrong answers can result in correct answers and this process needs only LLM-generated feedback and no other signals.
>
> > The authors primarily relied on accuracy metrics. However, due to the lack of calibration mitigation in the baselines, these metrics are influenced by label distribution.
>
> We additionally include F1, precision, recall on Knowledge Crosswords and COM2:
>
> |||**KC**||||**COM2**||||
> |--------------------|--------------------|------|------|------|------|------|------|------|------|
> |Data|Method|Precision (Micro)|Precision (Macro)|Recall (Macro)|F1 (Macro)|Precision (Micro)|Precision (Macro)|Recall (Macro)|F1 (Macro)|
> |Original||0.564|0.561|0.559|0.559|0.689|0.679|0.682|0.676|
> |Self Generator|Pairwise Filter|0.627|0.622|0.622|0.621|0.699|0.692|0.697|0.689|
> ||Score M50|0.597|0.598|0.591|0.592|0.675|0.663|0.666|0.659|
> ||Score M10|0.584|0.581|0.579|0.579|0.683|0.674|0.677|0.666|
> ||Oracle|0.580|0.576|0.578|0.576|0.700|0.688|0.689|0.682|
> |Mix Generator|Pairwise Filter|0.574|0.572|0.569|0.570|0.698|0.687|0.692|0.686|
> ||Score M50|0.590|0.592|0.587|0.588|0.674|0.658|0.665|0.659|
> ||Score M10|0.565|0.563|0.562|0.562|0.680|0.671|0.676|0.666|
> ||Oracle|0.581|0.579|0.580|0.577|0.699|0.684|0.691|0.684|
>
> Wrong-over-wrong alignment can yield better results too! For open-ended dataset, Bio Generation and NLGraph, there aren’t well-defined F1, precision or recall.
>
> > The baselines were limited to the original models, without comparison against some state-of-the-art (SOTA) alignment methods.
>
> Our work is a way to generate and select alignment data and not an alignment algorithm. That’s why DPO and more are orthogonal to the data-side improvements we are making in this work.
>
> While we employ DPO in the main paper, we additionally test out IPO, SPPO, ORPO in Figure 4, which suggests stable preference optimization methods like DPO are needed for wrong-over-wrong alignment.

---

> > ### Comment · Reviewer_AQ5T · 2024-11-18
> >
> > Thank you for the authors' response. The response has addressed some of my questions, but I still have some confusion regarding the generation method of wrong-over-wrong pairs in the paper.
> >
> > Intuitively, if answer A is less wrong than B, it is often also considered that B is more correct relative to A. When (A, B) is used as a pair to optimize the model, the optimization is based on the order B > A. Therefore, what is the core difference between wrong-over-wrong and right-over-wrong?

---

> ### Author Response · Authors · 2024-11-18
> **Official Comment by Authors**
>
> Right-over-wrong relies on **absolute** correctness to construct preference, where the chosen response is provided as the ground-truth in the dataset [1, 2]. Conversely, wrong-over-wrong alignment is grounded in a **spectrum** of wrongness, where preferences are constructed by choosing less wrong responses and rejecting more wrong ones. This enables alignment in cases where tasks are too complex or data-scarce to provide reliable ground-truths.
>
> [1] Wang, Peiyi, et al. "Making large language models better reasoners with alignment."
>
> [2] Chen, Nuo, et al. "GraphWiz: An Instruction-Following Language Model for Graph Problems."

---

> > ### Comment · Reviewer_AQ5T · 2024-11-22
> >
> > Thank you for the authors' reply; I have increased my rating.

---

### Official Review · Reviewer_LygQ · 2024-11-01

**Soundness:** 2
**Presentation:** 2
**Contribution:** 2
**Rating:** 6
**Confidence:** 4

**Summary:**

The paper explores utilizing LLMs to discern between incorrect answers i.e. wrong-over-wrong preferences and aligning LLMs using these preferences in situations where correct answers are unavailable or annotations are scarce.

**Strengths:**

- The paper explores a novel concept of "wrong-over-wrong" alignment, presenting an approach that diverges from traditional correct-answer evaluations, especially in contexts where correct answers may be absent.
- It raises important questions about how incorrect answers can still provide value in training and evaluation processes, potentially paving the way for further research in this area.

**Weaknesses:**

1. The framing of the concept could be improved to clearly communicate the utility of wrong-over-wrong alignment. The lack of explicit applications or contexts weakens the paper's stance. For instance, theorem-proving and low-resource languages are mentioned but the paper does not carry out any experiments on the same. Authors mentioned in the results that knowledge-based tasks had better performance but these tasks inherently contain correct answers which diverge from the proposed applications. Moreover, dataset curation inherently requires the ground truth to be present, which contradicts the motivation of the paper, which is that it can be deployed on tasks with scarce or no ground truths. The method performs poorly on SciBench which can be considered as a task close to theorem proving. Overall, the framing of concepts and applications can be improved.
2. The assessment of wrongness lacks rigorous criteria, leading to potential ambiguity in what constitutes "less wrong." More robust evaluation metrics are needed to substantiate claims. The subjectivity of wrongness varies drastically from task to task. The complexity of evaluating incorrect responses and their subjective interpretations requires more detailed explanations and examples to improve reader comprehension.
3. Currently the paper presents evaluations on different tasks and various parameters of its own method. But, it lacks sufficient comparison with existing models which are not aligned to wrong over wrong tasks.
4. The paper uses Logit-based methods to evaluate the correctness of a sample. They justify it by mentioning high token probability “probably” implies correct. However, models may assign high logits to incorrect answers due to biases in training data, leading to misleading assessments of accuracy[1].
5. The heuristic that longer answers equate to higher quality can lead to misjudgments, as verbosity may obscure clarity and accuracy, making it difficult to evaluate the substance of the response.  In [3], the authors evaluate the factuality of long-form responses, showing that simply increasing the length of an answer does not guarantee that it will contain accurate or relevant information. [4] emphasizes that longer outputs can often be less reliable due to various factors, including how models memorize factual knowledge.
6. Weak evaluation metrics: Given the small number of options in certain tasks, accuracy is a weak metric to fairly assess the performance, methods such as f1, precision, and recall provide more reliable results in such scenarios. ECE has limitations due to its reliance on binning, sensitivity to data distribution, and focus on average calibration. The choice of bins can introduce arbitrariness, affecting ECE values and leading to varying conclusions about a model’s calibration quality. Furthermore, ECE may be disproportionately impacted by bins with few data points, skewing results. As a summary statistic, it lacks detailed insights into calibration across specific confidence levels or types of inputs, making it less informative for models that need to distinguish varying degrees of incorrectness.
[1] https://proceedings.mlr.press/v162/wei22d.html
[3] https://arxiv.org/abs/2403.18802
[4] https://arxiv.org/abs/2212.10511

**Questions:**

1. Context of Application: Can the specific types of tasks that are most suitable for the wrong-over-wrong alignment approach be elaborated? This will help clarify the method's applicability and relevance.

2. The dataset creation process excludes correct answers and hence requires correct answers to be available. How does the paper reconcile this with the assertion that the method can be applied in contexts where ground truth is absent? A more detailed explanation is necessary.

3. Evaluation of Wrongness: What criteria were used to quantify the degree of wrongness in responses? Given the subjectivity involved, a detailed framework or methodology would enhance the validity of your evaluations.

4. The paper suggests that the model generates 4.5% more correct answers despite only aligning on wrong answers. What specific mechanisms can be hypothesized to contribute to this improvement? Further analysis could provide critical insights as it is noteworthy how without using correct answer information the model was able to perform better.

5. The paper evaluated the method on ScieBench which can be considered a task close to theorem proving. But, it does not perform up to the mark on the task. Can the paper discuss the implications of your findings for other datasets or tasks and as to why the method lacked generalizability to the task?

6. Evaluation Metrics: Can the paper focus why accuracy was chosen as the primary evaluation metric? Given the small number of options in certain tasks, would it be more appropriate to use additional metrics such as F1, precision, and recall to better capture model performance?

7. The paper mentions potential benefits for low-resource languages. Can specific examples or preliminary results demonstrate how the method could be applied effectively in this context?

8. While LLM as a judge holds promise, to use LLMs to judge themselves for something as subjective as wrongness with lack of ground truth should be justified with empirical results demonstrating the robustness of the evaluation. Can further experiments be performed to address this considering the paper compares different models as judges as part of their evaluations. Additionally, to use LLM as a judge, previous examples have to be given to align the LLM for evaluations. Can the paper mention how the examples have to be chosen for a given task and calibration of the LLM judges and how were they selected?

9. Pairwise evaluations - The evaluation results are mostly in the range of random answers even with filtering. Can the paper have a deeper analysis that justifies these results not being random?

10. There is a huge variance in metric performance with 1 out of 22 overall scores being above 70 which is mentioned as 20 points better than random guessing while 12 of these reported overall scores are within 5 points of random guessing. Can the paper delve further into this as there is no consistent pattern that can help users choose the appropriate metrics, evaluators and method in general for their applications.

---

> ### Author Response · Authors · 2024-11-15
> **Author Response (1/3)**
>
> We would like to thank the reviewer for their thoughtful comments and feedback.
>
> > Authors mentioned in the results that knowledge-based tasks had better performance but these tasks inherently contain correct answers which diverge from the proposed applications.
>
> Surely we can get ground-truths for knowledge-based tasks. However, as stated in Line 39, obtaining accurate annotations for these tasks is often **challenging**, as annotation processes can be costly, time-intensive, and susceptible to noise. To address these limitations, we explored learning from incorrect answers as an alternative approach, aiming to capture valuable insights from massive incorrect answers without solely relying on perfect annotations.
>
> > The assessment of wrongness lacks rigorous criteria, leading to potential ambiguity in what constitutes "less wrong." More robust evaluation metrics are needed to substantiate claims. The subjectivity of wrongness varies drastically from task to task. The complexity of evaluating incorrect responses and their subjective interpretations requires more detailed explanations and examples to improve reader comprehension.
>
> We agree that the proxies of wrongness we use might be imperfect. That’s why **we argue to use LLM-as-a-judge for generating wrong-over-wrong preference instead of directly using proxies**. Also, to have more accurate evaluation and mitigate the bias brought by a single proxy, we include a wide range of proxies across various tasks, as detailed in Section 3 and Appendix A and consistent improvements are observed.
>
> We provide explanations of each proxy we use in Section 3. Examples of what are less wrong answers for each proxy are in Table 14, Table 18, Table 20 and Table 22.
>
> > Currently the paper presents evaluations on different tasks and various parameters of its own method. But, it lacks sufficient comparison with existing models which are not aligned to wrong over wrong tasks.
>
> WoW-aligned models are compared to non-WoW aligned models in the first row of Table 2. We also compare wrong-over-wrong alignment with right-over-wrong alignment in Table 3, which suggests wrong-over-wrong alignment is a good supplement to right-over-wrong alignment.
>
> > The paper uses Logit-based methods to evaluate the correctness of a sample. They justify it by mentioning high token probability “probably” implies correct. However, models may assign high logits to incorrect answers due to biases in training data, leading to misleading assessments of accuracy.
>
> Logit-based method is just a baseline, and we also find it unreliable by experiment results in Table 1. That’s why we suggest using LLM-as–judge and is also validated by results in Table 1.
>
> > The heuristic that longer answers equate to higher quality can lead to misjudgments, as verbosity may obscure clarity and accuracy, making it difficult to evaluate the substance of the response.
>
> Length heuristics is just a baseline, and we also find it unreliable  by experiment results in Table 1. That’s why we suggest using LLM-as–judge and it is also validated by results in Table 1.
>
> > ECE has limitations due to its reliance on binning, sensitivity to data distribution, and focus on average calibration.
>
> ECE is the most commonly used calibration metric. [1,2,3,4,5] To better address your concern, will you please suggest any calibration metrics better than ECE? We are more than happy to report.
>
> [1] Geng, J., Cai, F., Wang, Y., Koeppl, H., Nakov, P., & Gurevych, I. (2024, June). A Survey of Confidence Estimation and Calibration in Large Language Models. In Proceedings of the 2024 Conference of the North American Chapter of the Association for Computational Linguistics: Human Language Technologies (Volume 1: Long Papers)
>
> [2] Shangbin Feng, Weijia Shi, Yike Wang, Wenxuan Ding, Vidhisha Balachandran, and Yulia Tsvetkov. 2024. Don’t Hallucinate, Abstain: Identifying LLM Knowledge Gaps via Multi-LLM Collaboration. In Proceedings of the 62nd Annual Meeting of the Association for Computational Linguistics (Volume 1: Long Papers)
>
> [3] Jiang, Z., Araki, J., Ding, H., & Neubig, G. (2021). How can we know when language models know? on the calibration of language models for question answering. Transactions of the Association for Computational Linguistics, 9, 962-977.
>
> [4] Tao, L., Zhu, Y., Guo, H., Dong, M., & Xu, C. A Benchmark Study on Calibration. In The Twelfth International Conference on Learning Representations.
>
> [5] Liu, X., Khalifa, M., & Wang, L. (2024). LitCab: Lightweight Language Model Calibration over Short-and Long-form Responses. In The Twelfth International Conference on Learning Representations.

---

> ### Author Response · Authors · 2024-11-15
> **Author Response (2/3)**
>
> > Context of Application: Can the specific types of tasks that are most suitable for the wrong-over-wrong alignment approach be elaborated? This will help clarify the method's applicability and relevance.
>
> > The paper mentions potential benefits for low-resource languages. Can specific examples or preliminary results demonstrate how the method could be applied effectively in this context?
>
> While theorem-proving and multilingual tasks serve as motivating examples for this paper, they lack reliable automated evaluation metrics (e.g. how to automatically determine if a theorem proof is correct or not?), which led us to focus on Knowledge Crosswords and COM2 instead. We revise Line 41 to add "structured reasoning” as an application.
>
> In fact, Knowledge Crosswords and COM2 tasks are highly challenging and require costly annotations due to their complex reasoning structure and the vast range of possible responses, which makes manual verification both difficult and labor-intensive. Improvements on these tasks using wrong-over-wrong alignment, demonstrating its utility in handling scenarios without ground-truths.
>
> > Moreover, dataset curation inherently requires the ground truth to be present, which contradicts the motivation of the paper, which is that it can be deployed on tasks with scarce or no ground truths.
>
> >The dataset creation process excludes correct answers and hence requires correct answers to be available. How does the paper reconcile this with the assertion that the method can be applied in contexts where ground truth is absent? A more detailed explanation is necessary.
>
> As stated from line 172 to line 173, using ground-truth to filter out correct answers only aims to study model behavior under the worst case when there are no correct answers at all. It’s totally unnecessary to avoid right-over-wrong alignment in practice. In fact, we prove in Section 5 that wrong-ovre-wrong alignment is a good supplement to right-over-wrong alignment. Also, in the footnote of page 2, we state we employ datasets with ground-truths but only for evaluation purposes.
>
> > Evaluation of Wrongness: What criteria were used to quantify the degree of wrongness in responses? Given the subjectivity involved, a detailed framework or methodology would enhance the validity of your evaluations.
>
> As stated in Section 3, we use proxy functions to evaluate the wrongness of answers. For example,  to find the shortest path in a network with a ground truth path length of 5, finding a path of length 8 is “less wrong” than a path of length 11. [1] Each proxy is carefully designed to suit their unique nature and thus we believe it is unnecessary to unify the evaluation of wrongness into a general framework. Also, to have more accurate evaluation and mitigate the subjectivity brought by a single proxy, we include a wide range of proxies across various tasks, as detailed in Section 3 and Appendix A and consistent improvements are observed.
>
> We will also consider using the proxy weighting proposed by Reviewer aJUG in future work to have more robust evaluation.
>
> > The paper suggests that the model generates 4.5% more correct answers despite only aligning on wrong answers. What specific mechanisms can be hypothesized to contribute to this improvement? Further analysis could provide critical insights as it is noteworthy how without using correct answer information the model was able to perform better.
>
> We contribute “aligning with wrong to get right” to the distribution shift from Line 197 to Line 201 in Section 2.2. We believe wrongness is a spectrum. As the model refines its wrong-over-wrong separation, the generation distribution may move towards “less wrong” direction. Thus answers that were previously close-to-correct may be adjusted enough to align with the correct solution, even though the training process only employs incorrect answers.
>
> > The method performs poorly on SciBench which can be considered as a task close to theorem proving. Overall, the framing of concepts and applications can be improved.
>
> SciBench is by no means close to theorem proving, since there are no clear wrongness distinctions among LLM-generated wrong answers. We experiment on SciBench only to discuss wrong-over-wrong alignment may not apply to all domains, especially those without clear wrongness distinctions.
>
>
> [1] Wang, H., Feng, S., He, T., Tan, Z., Han, X., & Tsvetkov, Y. (2023). Can language models solve graph problems in natural language?. Advances in Neural Information Processing Systems, 36.

---

> ### Author Response · Authors · 2024-11-15
> **Author Response (3/3)**
>
> > Evaluation Metrics: Can the paper focus why accuracy was chosen as the primary evaluation metric? Given the small number of options in certain tasks, would it be more appropriate to use additional metrics such as F1, precision, and recall to better capture model performance?
>
> We additionally include F1, precision, recall on Knowledge Crosswords and COM2:
>
> |||**KC**||||**COM2**||||
> |--------------------|--------------------|------|------|------|------|------|------|------|------|
> |Data|Method|Precision (Micro)|Precision (Macro)|Recall (Macro)|F1 (Macro)|Precision (Micro)|Precision (Macro)|Recall (Macro)|F1 (Macro)|
> |Original||0.564|0.561|0.559|0.559|0.689|0.679|0.682|0.676|
> |Self Generator|Pairwise Filter|0.627|0.622|0.622|0.621|0.699|0.692|0.697|0.689|
> ||Score M50|0.597|0.598|0.591|0.592|0.675|0.663|0.666|0.659|
> ||Score M10|0.584|0.581|0.579|0.579|0.683|0.674|0.677|0.666|
> ||Oracle|0.580|0.576|0.578|0.576|0.700|0.688|0.689|0.682|
> |Mix Generator|Pairwise Filter|0.574|0.572|0.569|0.570|0.698|0.687|0.692|0.686|
> ||Score M50|0.590|0.592|0.587|0.588|0.674|0.658|0.665|0.659|
> ||Score M10|0.565|0.563|0.562|0.562|0.680|0.671|0.676|0.666|
> ||Oracle|0.581|0.579|0.580|0.577|0.699|0.684|0.691|0.684|
>
> Wrong-over-wrong alignment can yield better results too! For open-ended dataset, Bio Generation and NLGraph, there aren’t well-defined F1, precision or recall.
>
> > While LLM as a judge holds promise, to use LLMs to judge themselves for something as subjective as wrongness with lack of ground truth should be justified with empirical results demonstrating the robustness of the evaluation. Can further experiments be performed to address this considering the paper compares different models as judges as part of their evaluations. Additionally, to use LLM as a judge, previous examples have to be given to align the LLM for evaluations. Can the paper mention how the examples have to be chosen for a given task and calibration of the LLM judges and how were they selected?
>
> That’s just the motivation of the first experiment in Table 1, we use proxies to validate LLM’s capability of judging wrongness. To have more accurate evaluation and mitigate the subjectivity brought by a single proxy, we include a wide range of proxies across various tasks, as detailed in Section 3 and Appendix A and consistent improvements are observed.
>
> Do you mean “few-shots examples” when you say “previous examples”? We believe “to use LLM as a judge, previous examples have to be given to align the LLM for evaluations.” is overclaimed. Some previous works have explored LLM-as-a-judge without few-shot examples, like [1]. The prompt we used to guide LLM-as-a-judge is shown in Table   12 and Table 13 and no “previous examples” are used.
>
> > Pairwise evaluations - The evaluation results are mostly in the range of random answers even with filtering. Can the paper have a deeper analysis that justifies these results not being random?
>
> In Table 1, we aim to find out the best way to elicit wrong-over-wrong preferences. From Line 305 to line 354, we discuss that methods such as heuristic, consistency-based, logits-based, indeed perform close to random guess. That’s why we suggest using LLM-as-a-judge (pairwise comparison and score-based) with filtering which generates clearly not random wrong-over-wrong preferences.
>
> > There is a huge variance in metric performance with 1 out of 22 overall scores being above 70 which is mentioned as 20 points better than random guessing while 12 of these reported overall scores are within 5 points of random guessing. Can the paper delve further into this as there is no consistent pattern that can help users choose the appropriate metrics, evaluators and method in general for their applications.
>
> We believe this paper already points out the best method to elicit wrong-over-wrong preferences from Line 305 to Line 309. In Table 1, LLM-as-a-judge (pairwise comparison and score-based) with filtering generates clearly reasonably good wrong-over-wrong preferences. Besides, we also discuss the preference accuracy and base model capability in Figure 2, which suggests using larger LMs is better for eliciting wrong-over-wrong preferences.
>
> [1] Zeng, Z., Yu, J., Gao, T., Meng, Y., Goyal, T., & Chen, D. (2023). Evaluating large language models at evaluating instruction following. arXiv preprint arXiv:2310.07641.

---

> ### Author Response · Authors · 2024-11-25
> **Author Response**
>
> We are thankful for your constructive comments and feedback: we have incorporated all your suggested edits and posted an updated version. The updates include but are not limited to presenting motivation more clearly, clarifying methodology and experiment design, adding new experiments and results, providing more analysis and discussion on results and fixing typos. We would appreciate it if you might have any further feedback.
>
> Thank you, authors

---

> > ### Comment · Reviewer_LygQ · 2024-11-25
> >
> > Thanks for your detailed responses. Many things are clarified now. Please make sure to include these points in the updated draft. I have increased my score.

---

### Official Review · Reviewer_UTTj · 2024-11-02

**Soundness:** 3
**Presentation:** 3
**Contribution:** 3
**Rating:** 6
**Confidence:** 3

**Summary:**

The paper explores aligning LLMs with "wrong-over-wrong" preferences, a novel approach to distinguishing varying degrees of incorrectness rather than simply selecting between correct and incorrect answers. This technique aims to enhance not only the probability assigned to "less wrong" responses but also improves model accuracy and calibration.

**Strengths:**

* Creative approach that addresses alignment without needing correct answers.
* Solid, methodologically rigorous results across diverse datasets, validated with multiple LLMs.

**Weaknesses:**

* The results and analysis sections are presented in a "bullet point" format without logical transitions, making it feel like a collection of findings rather than a cohesive story.
* Most tables, especially Table 1, are dense and challenging to interpret. Consider breaking them down, or move detailed tables to an appendix and keep summary statistics or visualizations.
* In the experimental settings section, more context around the overall setup would help; for instance, the mention of multiple-choice questions jumps in without prior explanation.

**Questions:**

* Why not average over evaluators? The paper doesn't focus on LLM-as-a-judge, so differentiating evaluators doesn’t seem essential in this context.
* The claim that "knowledge-based tasks are easier while commonsense is most challenging" feels overstated given the relatively modest performance gap.

**Minor Comments:**
* I would appreciate a clearer motivation, especially in the abstract and introduction; perhaps clarify how this method could address practical limitations in LLMs.

---

> ### Author Response · Authors · 2024-11-15
> **Author Response**
>
> We would like to thank the reviewer for their thoughtful comments and feedback.
> > The results and analysis sections are presented in a "bullet point" format without logical transitions, making it feel like a collection of findings rather than a cohesive story.
>
> > The claim that "knowledge-based tasks are easier while commonsense is most challenging" feels overstated given the relatively modest performance gap.
>
> We rearrange the order and name of each “bullet point” to have a clear logical flow. For example, the bullet points of  first phase experiments are:
>
> + Overall: Feasible to elicit wrong-over-wrong preference.
> + Best eliciting method: Scored-based.
> + Improve upon original eliciting method: consistency checks and score margins.
> + Failed eliciting mode: self-evaluation.
>
> We do that for the second phase and analysis experiment too. Moreover, we change the bullet point “knowledge-based tasks are easier while commonsense is most challenging" to “knowledge-based tasks showed a performance advantage in our tests. However, this difference is not significant and may not apply universally.” and put it in the Appendix A due to lack of significance.
>
> > Most tables, especially Table 1, are dense and challenging to interpret. Consider breaking them down, or move detailed tables to an appendix and keep summary statistics or visualizations.
>
> We merge Table 1 and Table 9 and move it to Appendix A. We only keep the “evaluator-independent” and “GPT-4o as evaluator” part of Table 1 in the main page. We also visualize the “overall” column as a bar chart.
>
> > In the experimental settings section, more context around the overall setup would help; for instance, the mention of multiple-choice questions jumps in without prior explanation.
>
> Due to space limitation, detailed experiment setup is moved to Appendix B. We add a sentence “More detailed experiment setup can be found in Appendix B” in Line 219 and introduce the definition of multiple-choice questions in Appendix B.
>
> > Why not average over evaluators? The paper doesn't focus on LLM-as-a-judge, so differentiating evaluators doesn’t seem essential in this context.
>
> We acknowledge that averaging evaluators could simplify the analysis. We chose to report results separately for each evaluator to investigate consistency across LLMs and understand potential evaluator effects, such as LLMs may introduce unique biases when judging their own generation[1, 2], which may offer insight for evaluator selection.
>
> We add an averaged performance table over evaluators in the appendix A to summarize these findings for readers who prefer a consolidated view.
>
> > I would appreciate a clearer motivation, especially in the abstract and introduction; perhaps clarify how this method could address practical limitations in LLMs.
>
> We add "structured reasoning” as an application in Line 41. We also add “This paper looks at the massive low-quality or even completely wrong answers generated by LLMs which are overlooked by previous work, and investigate under the worst scenario where there are no ground-truths available, can we still expand the frontier of model capabilities using those low-qaulity answers.” in Line 45.
>
> [1] Valmeekam, K., Marquez, M., & Kambhampati, S. (2023). Can large language models really improve by self-critiquing their own plans?. arXiv preprint arXiv:2310.08118.
>
> [2] West, P., Lu, X., Dziri, N., Brahman, F., Li, L., Hwang, J. D., ... & Choi, Y. (2023, October). THE GENERATIVE AI PARADOX:“What It Can Create, It May Not Understand”. In The Twelfth International Conference on Learning Representations.

---

### Official Review · Reviewer_aJUG · 2024-11-03

**Soundness:** 3
**Presentation:** 3
**Contribution:** 3
**Rating:** 8
**Confidence:** 4

**Summary:**

This paper presents a creative approach to solving alignment problem in large language models (LLMs) especially in resource constraint settings by proposing a methodology called "wrong-over-wrong alignment." The core idea is for LLMs to learn by differentiating varying degrees of wrongness in responses, enabling alignment without the need for correct, ground-truth answers. This approach addresses a common challenge in real-world applications where human-verified data is often scarce. By relying on wrongness comparisons, the authors demonstrate that this alignment can enhance model calibration and improve response accuracy even in resource-limited settings.

**Strengths:**

This paper introduces quite a novel approach to doing alignment in the absence of ground truth. The paper proposes an alignment methodology called wrong-over-wrong alignment. Such an alignment allows any LLM to learn the correctness of an answer (for the domain it is trained/aligned on), by simply distinguishing between varying shades of wrong. This is quite a valuable approach to solve alignment especially when human vetted groudth truth is missing, which is more often than not in any practical situation. This idea has the potential to push the capabilities of LLMs in resource constrained settings.

Authors have done a good job in presenting a very comprehensive, clear and rigorous experimentation methodology to test the hypothesis. I was quite satisfied to read about various methods that the authors proposed to elicit wrong over wrong preferences. I also came out quite impressed with the experimentation rigor presented in the paper. From various preference elicitation methods to various LLMs, authors have ensured sufficient dimensions are explored in depth.

Lastly, various insights such as task utility, effectiveness of wrong over wrong preferences, innate capabilities of LLMs, model confidences etc strengthened the paper further.

**Weaknesses:**

The biggest weakness in my opinion of the method presented in tihs paper is its reliance on proxy functions. While the main objective of the paper is to explore alignment without dependence on correct answers, the paper however relies on proxy functions to elicit wrongness of answers. The proxy methods come with their own limitations and I worry that in real world practice, they may inadvertently introduce biases or even inaccuracies. Authors themselves have called out a possible introduction of biases in domains where the proxies are not well represented. Having said this, its not clear what unbiased alternative proxy methods one could use or even do away with proxies.
As an ML practioner in an industrial setting, I would have liked to see experiments on real world use cases where implicit signals such as lack of clicks (normalized according to position) act as proxies for assigning wrongness score. The paper has an opportunity to test this idea in wild and use real world implicit signals to elicit wrongness. There is more wrongness signal than correctness signal in any real world application and thus this idea has huge potential.

**Questions:**

My main question to author is how confident they are in proxies' ability to accurately represent wrongness across different tasks. Have authors been thinking about doing proxy weighting to further improve their reliability.

 On the insights, another question that I would like to get answers on is if authors observe any patterns in failure cases ? I am looking for cases where wrong over wrong alignment did not produce reliable preferences. What were those patterns ?

I mentioned this in above section too. I am curious to learn how well does wrong over wrong alignment generalize to out of distribution tasks. Have the authors tested whether the model's calibration is preserved across tasks ?

How do the authors plan to use this method in more subjective domains ? We get into the ethical contexts where defining what is less wrong could be culturally vary. This, in some sense, could be tied back to domain specific proxy definitions ? What are authors' thoughts and what guidance they want to provide to the community wanting to use this method in practice.

---

> ### Author Response · Authors · 2024-11-15
> **Author Response (1/2)**
>
> We would like to thank the reviewer for their thoughtful comments and feedback.
>
> > While the main objective of the paper is to explore alignment without dependence on correct answers, the paper however relies on proxy functions to elicit wrongness of answers.
>
> The core thing is: **wrong-over-wrong preferences are elicited from LLMs and not with proxy scores**. Proxy scores are employed to evaluate LLM-generated preferences: they only serve for evaluation purposes as stated in the footnote of Page 2.
>
> The process essentially is: take a question, LLM sample multiple answers, LLM self-judge wrongness preferences among answers, align with these preferences, and then evaluate with the proxy function. In other words, aligning with only wrong answers can result in correct answers and this process needs only LLM-generated feedback and no other signals.
>
> > Proxies may inadvertently introduce biases or even inaccuracies. I would have liked to see experiments on real world use cases where implicit signals such as lack of clicks (normalized according to position) act as proxies for assigning wrongness score.
>
> Proxies may be imperfect and that’s why we use LLM-as-a-judge to generate wrong-or-wrong preferences instead of directly using proxies. Also, to have more accurate evaluation and mitigate the bias brought by a single proxy, we include a wide range of proxies across various tasks, as detailed in Section 3 and Appendix A and we see consistent improvements across all employed proxies.
>
> While our current scope focuses on controlled datasets to isolate and rigorously evaluate wrong-over-wrong alignment, we are inspired by the suggestion to incorporate implicit real-world signals (such as click through rate) as proxies, particularly for subjective or practical applications. This will be a valuable addition for validating the robustness of our method in future work.
>
> > My main question to author is how confident they are in proxies' ability to accurately represent wrongness across different tasks. Have authors been thinking about doing proxy weighting to further improve their reliability?
>
> We believe the proxies we use, although not perfect, are nonetheless reasonable. A failure case could be that some proxies stress much on the wrongness of the final answer but overlook the hallucination in reasoning steps. For example, in NLGraph, an answer finding a shortest path of 11 may have correct calculation while an answer giving 8 may come from wrong calculation (the right answer is 5). But overall manual examination seems good as shown in Table 22.
>
> We appreciate the proxy weighting idea, which can improve reliability in areas where robust proxies are limited in future work. However, for the datasets we experimented with, a single proxy may be sufficient, as confirmed by the results.
>
> > On the insights, another question that I would like to get answers on is if authors observe any patterns in failure cases ? I am looking for cases where wrong over wrong alignment did not produce reliable preferences. What were those patterns?
>
> We provide failure examples and analysis of 4 datasets in Table 16, Table 19, Table 21, and Table 24 in the appendix. The failed patterns we observed are similar to that before wrong-over-wrong alignment but appear less frequently. This validates our assumption that the benefit of wrong-over-wrong alignment comes from distribution shift as discussed in Section 2.2.

---

> > ### Comment · Reviewer_aJUG · 2024-11-24
> >
> > Thank you for the responses to my question. I also read the responses to other reviewers' comments. I will keep my current rating so far.

---

> ### Author Response · Authors · 2024-11-15
> **Author Response (2/2)**
>
> > I mentioned this in above section too. I am curious to learn how well does wrong over wrong alignment generalize to out of distribution tasks. Have the authors tested whether the model's calibration is preserved across tasks?
>
> The generalization of wrong-over-wrong alignment can be found in Table 5 and Section 5. It is noticeable that wrong-over-wrong alignment generalizes well on in-domain but unseen tasks on less wrong, more correct, and better calibration.
>
> We agree that it is unclear whether wrong-over-wrong alignment can generalize to out-of-domain tasks due to the nature of post-training, which could make LLM better at certain tasks but harm general performance.
>
> > How do the authors plan to use this method in more subjective domains ? We get into the ethical contexts where defining what is less wrong could be culturally vary. This, in some sense, could be tied back to domain specific proxy definitions ? What are authors' thoughts and what guidance they want to provide to the community wanting to use this method in practice.
>
> As discussed in the ethical statement, we are concerned that wrong-over-wrong alignment may be misused in subjective context to justify some unsafe responses as “less wrong” or “more acceptable”, while they should all be avoided.
>
> However, with proper supervision, wrong-over-wrong alignment can effectively support personalization. For example, pluralistic alignment [1] may struggle with insufficient high-quality golden generations (ground-truths) for low-resource demographic groups. We can collect LLMs’ generation from high-resource demographic groups and apply wrong-over-wrong alignment on preferences obtained from low-resource demographic groups.
>
> [1] Sorensen, T., Moore, J., Fisher, J., Gordon, M. L., Mireshghallah, N., Rytting, C. M., ... & Choi, Y. Position: A Roadmap to Pluralistic Alignment. In the Forty-first International Conference on Machine Learning.

---

### Meta-Review · Area_Chair_5tuz · 2024-12-21

**Metareview:**

This paper introduces a novel solution to the alignment problem in large language models (LLMs), particularly for resource-constrained scenarios, through a method called "wrong-over-wrong alignment." The key concept involves training LLMs to distinguish between varying levels of incorrectness in responses, bypassing the need for correct, ground-truth answers. This approach tackles a prevalent challenge in real-world applications where human-verified data is scarce. By leveraging comparisons of wrongness, the authors show that this method improves model calibration and enhances response accuracy, even in settings with limited resources.

Positive points:
+ The idea proposed in the paper is interesting
+ The model is solid, methodologically rigorous
+ The experiments can well demonstrate the effectiveness of the model.

Negative points:
- The proxy functions can be not reliable.
- More analysis on the experimental results.

**Additional Comments On Reviewer Discussion:**

In the rebuttal period, the authors provided very detailed responses to the reviewers' comments. In the responses, I think most concerns initially raised by the reviewers have been addressed. At last, all the reviewers give positive ratings. I recommend acceptance of the paper.

---

### Decision · Program_Chairs · 2025-01-22

Accept (Poster)